# ResPrompt: Residual Connection Prompting Advances Multi-Step Reasoning in Large Language Models

## Abstract

Chain-of-thought (CoT) prompting, which offers step-by-step problem-solving rationales, has impressively unlocked the reasoning potential of large language models (LLMs). Yet, the standard CoT is less effective in problems demanding multiple reasoning steps. This limitation arises from the complex reasoning process in multi-step problems: later stages often depend on the results of several steps earlier, not just the results of the immediately preceding step. Such complexities suggest the reasoning process is naturally represented as a *graph*. The almost *linear and straightforward* structure of CoT prompting, however, struggles to capture this complex reasoning graph. To address this challenge, we propose *Residual Connection Prompting* (ResPrompt), a new prompting strategy that advances multi-step reasoning in LLMs. Our key idea is to reconstruct the reasoning graph within prompts. We achieve this by integrating necessary connections—links present in the reasoning graph but missing in the linear CoT flow—into the prompts. Termed "*residual connections*", these links are pivotal in morphing the linear CoT structure into a graph representation, effectively capturing the complex reasoning graphs inherent in multi-step problems. We evaluate ResPrompt on six benchmarks across three diverse domains: math, sequential, and commonsense reasoning. For the open-sourced LLaMA family of models, ResPrompt yields a significant average reasoning accuracy improvement of 12.5% on LLaMA-65B and 6.8% on LLaMA2-70B. Breakdown analysis further highlights ResPrompt particularly excels in complex multi-step reasoning: for questions demanding at least five reasoning steps, ResPrompt outperforms the best CoT based benchmarks by a remarkable average improvement of 21.1% on LLaMA-65B and 14.3% on LLaMA2-70B. Through extensive ablation studies and analyses, we pinpoint how to most effectively build residual connections, and assess ResPrompt in view of "emergent ability", few-shot learning, and robustness, while also noting scenarios in which it might be superfluous.

## 1 Introduction

Recent advancements in scaling up large language models (LLMs) (Brown et al., 2020; Thoppilan et al., 2022; Chowdhery et al., 2022; Anil et al., 2023; Touvron et al., 2023a;b; Zeng et al., 2023; Scao et al., 2022; Zhang et al., 2022; Zhao et al., 2023; Yang et al., 2023) have not only significantly improved their performance but have also enabled entirely new "emergent ability" (Wei et al., 2022a). One milestone approach that harnesses this potential is

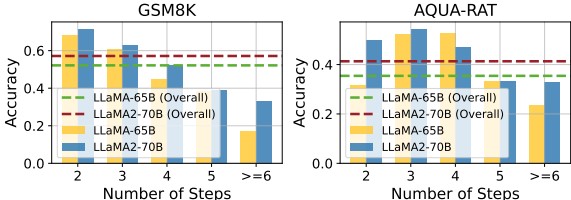

Figure 1: CoT reasoning accuracy based on the number of reasoning steps for LLaMA-65B and LLaMA2-70B across two math benchmarks. Horizontal dashed lines are the overall accuracy in each benchmark. Left: GSM8K, 8-shot; Right: AQUA-RAT, 4-shot. CoT prompts are sourced from (Wei et al., 2022b).

chain-of-thought (CoT) prompting (Wei et al., 2022b), which uses few-shot step-by-step demonstrations to teach LLMs how to reach a final answer. CoT prompting has unlocked impressive reasoning abilities in LLMs, enabling them to excel in various complex tasks, including mathematics, commonsense reasoning, logic, and more (Wei et al., 2022b; Suzgun et al., 2022; Lu et al., 2022).

However, standard CoT approach has proven to be less effective in addressing questions that involve multiple reasoning steps (Fu et al., 2023b; Zhou et al., 2023a; Khot et al., 2023). In Figure 1, we demonstrate that both LLaMA-65B (Touvron et al., 2023a) and LLaMA2-70B (Touvron et al., 2023b) experience a notable decline in performance as the number of reasoning steps increases on the mathematical benchmarks GSM8K (Cobbe et al., 2021) and AQUA-RAT (Ling et al., 2017).

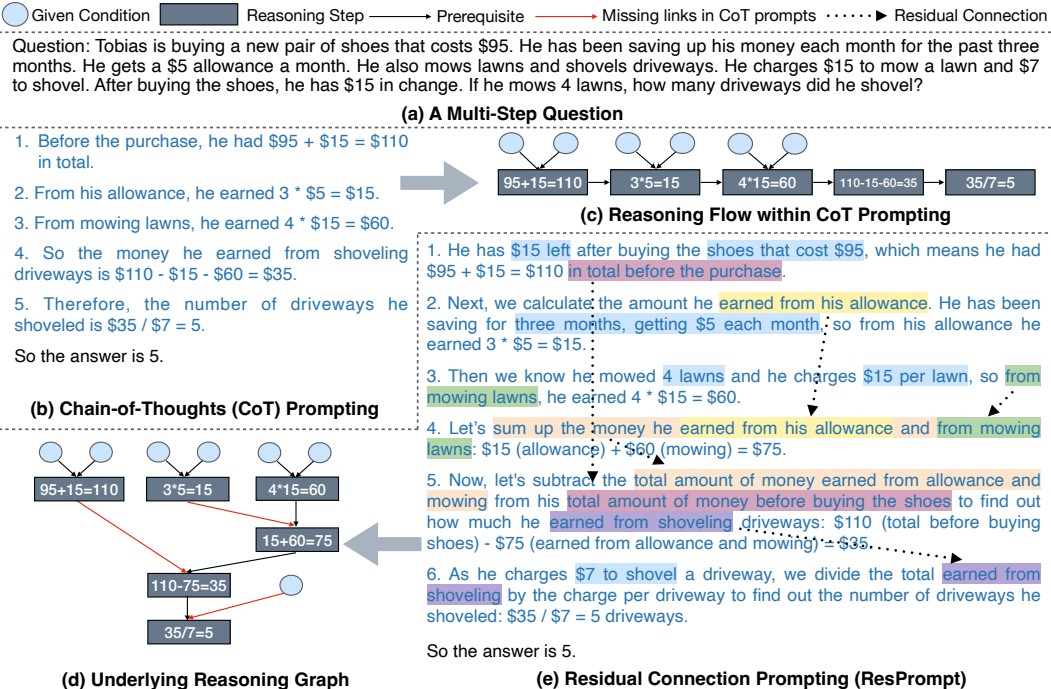

Figure 2: **(a)** A multi-step math question from the training set of GSM8K (Cobbe et al., 2021). **(b)** Standard CoT prompting for this question. The intermediate steps are highlighted in blue. **(c)** The reasoning flow within the CoT prompts in (b), which exhibits a linear structure. **(d)** The underlying complex reasoning graph of this math question. **(e)** Our approach, RESPROMPT (residual connection prompting) for this question. The intermediate steps are highlighted in blue, while residual connections are indicated with colored backgrounds and linked by dashed arrows. Note that phrases with a blue background represent given conditions from the question, while phrases with backgrounds in other colors denote results derived from intermediate steps.

Why is this the case? We hypothesize that in many multi-step reasoning processes, later stages rely not only on the immediately preceding step but also on results from *several steps prior* as prerequisites. This complex interdependence leads to the reasoning process in these multi-step questions essentially forming a graph structure, which we refer to as "*reasoning graph*". We show an example involving multi-step reasoning from GSM8K benchmark in Figure 2 (a) and its complex underlying reasoning graph in Figure 2 (d). However, the "step-by-step" nature of standard CoT prompts typically generates a nearly linear reasoning flow (see Figure 2 (b)-(c)). This simplistic reasoning flow within CoT prompts has a structural mismatch with the complex underlying reasoning graph, thereby limiting CoT's effectiveness in handling questions that require multiple reasoning steps[1].

To tackle this challenge, we propose *Residual Connection Prompting* (RESPROMPT), a new prompting strategy that bridges this structural gap in reasoning processes and thereby enhances the multi-step reasoning capability of LLMs. The core idea of RESPROMPT is to reconstruct the reasoning graph in prompts from the linearly structured reasoning flow via adding necessary connections. A necessary connection is a link present in reasoning graph but missing in linear reasoning flow (see red arrows in Figure 2 (d) for examples). Specifically, a necessary connection usually embodies the essential prerequisites of a reasoning step. Here, a prerequisite can be either an intermediate result

---

[1]It's worth noting that some multi-step questions may exhibit a simpler, nearly linear underlying reasoning flow where CoT prompts suffice. We dive into this aspect in detail in Section 3.5.

generated by earlier steps or a known condition from the question itself. In RESPROMPT, we explicitly link these prerequisites to their corresponding reasoning step by repeating them, using the same tokens, within that specific step in the prompts. By doing so, we effectively recover the complex underlying reasoning graphs of multi-step questions in the prompts of RESPROMPT. In Figure 2 (e), we present an example of RESPROMPT applied to a multi-step question. We call these explicit links as "*residual connections*" within prompts. This terminology is inspired by the residual connections across neural network layers (He et al., 2016). However, a critical distinction lies in the context-specific nature of residual connections in RESPROMPT. While the residual connections in He et al. (2016) are uniform, RESPROMPT's connections depend on the unique context, as prerequisites for each reasoning step might come from different positions in the reasoning process.

We use the open-sourced LLaMA family of models (LLaMA, LLaMA2) (Touvron et al., 2023a;b) to evaluate RESPROMPT on six benchmarks, including 1) Mathematical reasoning: GSM8K (Cobbe et al., 2021), AQUA-RAT (Ling et al., 2017), MathQA (Amini et al., 2019), SVAMP (Patel et al., 2021); 2) Sequential reasoning: SCONE-Alchemy (Long et al., 2016); and 3) Commonsense reasoning: StrategyQA (Geva et al., 2021). Our experiments demonstrate that RESPROMPT significantly improves overall reasoning accuracy by an average of 12.5% on LLaMA-65B and 6.8% on LLaMA2-70B. In particular, breakdown analysis shows our performance gains on multi-step questions are much more remarkable: for questions requiring at least 5 reasoning steps, RESPROMPT outperforms the best CoT based approaches by an average improvement of 21.1% on LLaMA-65B and 14.3% on LLaMA2-70B. Furthermore, through extensive ablation studies and analyses, we investigate how to build residual connections most effectively. We dive into how RESPROMPT functions in terms of "emergent ability" (Wei et al., 2022a), few-shot learning, robustness, and conduct error analyses. Additionally, we discuss when RESPROMPT may not be necessary.

Note that unlike recent studies that organize steps into advanced structures such as trees (Yao et al., 2023a; Long, 2023) and graphs (Besta et al., 2023), which enable schemas like search and backtracking for strategic tasks, RESPROMPT focuses on capturing necesssary dependencies for reasoning steps to advance multi-step reasoning, making it widely applicable across various reasoning tasks.

## 2 RESPROMPT: RESIDUAL CONNECTION PROMPTING

### 2.1 WHY IS STANDARD COT LESS EFFECTIVE FOR MULTI-STEP REASONING?

To investigate the reasons for the failure of the standard CoT in multi-step reasoning, we use mathematical reasoning as our illustrative example. In Figure 2 (a), we present a math question from GSM8K (Cobbe et al., 2021), which consists of multiple reasoning steps. Note that in GSM8K, a step is annotated as one math calculation. However, this notion can also encompass similar ideas depending on the specific context (Fu et al., 2023b), such as a sub-question (Zhou et al., 2023a).

As shown in Figure 2 (d), a multi-step question exhibits a complex, structured underlying reasoning process, where later stages steps frequently depend not only on the immediately preceding step but also potentially on results *several steps prior*. This complex interdependence renders the underlying structure of reasoning flow essentially a graph, which we refer to as a *reasoning graph*. However, in standard CoT prompts, reasoning unfolds in a step-by-step manner, including only the immediately preceding step, with no explicit reference to intermediate results from several steps prior (Figure 2 (b)). This consequently yields a nearly linear-structured reasoning flow within the standard CoT prompts (Figure 2 (c)), which is not able to fully recover the complex underlying reasoning graphs inherent in multi-step questions. This striking mismatch in reasoning flow structures significantly impairs CoT's capacity to effectively tackle multi-step reasoning.

We note that while we use math problems as our running example in Figure 2, these findings are broadly applicable to any other types of multi-step problems characterized by complex reasoning flows. It's important to mention that not every multi-step question exhibits a graph-like reasoning process; some questions may involve a long chain of dependencies, which we explore in Section 3.5.

### 2.2 ENABLING MULTI-STEP REASONING VIA BUILDING RESIDUAL CONNECTIONS

**Principle and Methodology.** Our findings lead to the hypothesis that standard CoT struggles with multi-step reasoning because its nearly linear reasoning flow within prompts is not sufficient for

capturing the reasoning graphs inherent in complex multi-step questions. In a graphical view, the CoT reasoning flow, as shown in Figure 2 (c), misses necessary connections required to reconstruct the complex reasoning graph depicted in Figure 2 (d). A more intuitive interpretation is that CoT tends to "forget" intermediate results it has previously derived. To address this structural mismatch, we propose a novel prompting strategy aimed at reconstructing the complex underlying reasoning graph by explicitly adding the vital missing connections. These added connections re-introduce intermediate results from previous steps as prerequisites for later steps. Specifically, for a step, we first 1) *enumerate and connect the necessary prerequisites with either results of earlier steps or directly from the provided question conditions*, then we 2) *derive the result based on these prerequisites*. An example is shown in Figure 2 (e). We refer to our added links as "*Residual Connections*" and call our prompting strategy as *Residual Connection Prompting* (RESPROMPT). By building residual connections to recall essential prerequisites, RESPROMPT ensures that the reasoning flow within prompts sufficiently align with the underlying reasoning graphs for complex multi-step questions.

A natural question arises: where should we build residual connections for effective alignment with complex reasoning graphs in multi-step problems? Is it essential to introduce them at every single step, or would a selective subset suffice? We investigate this in ablation studies on residual connection placement in Section 3.3. Our findings emphasize that covering the entire reasoning process with residual connections is crucial for RESPROMPT's improved multi-step reasoning performance.

**Implementation.** In RESPROMPT, we build residual connections through a straightforward method: reuse the exact same tokens as references. That is, when recalling an intermediate result from a prior step, we describe it by repeating the exact same tokens. For example, in Figure 2 (e), we derive the phrase "earned from his allowance" (highlighted in yellow background) in the second step. To reference it as a prerequisite for the fourth step, we repeat "earned from his allowance" verbatim, facilitating LLMs in easily connecting the current step with prior intermediate results. In Section 3.3, we compare this approach with more efficient designs, such as representing intermediate results as a symbolic variable denoted as X and later directly reusing X. Our findings confirm that our straightforward exact repeat approach is more effective in building residual connections within prompts.

**Insights and Understanding.** RESPROMPT is a simple and effective approach. Our intuitive understanding regarding its strong performance in multi-step reasoning can be distilled into two key perspectives: 1) *Recovering complex reasoning graphs*. As previously discussed, residual connections play a crucial role in sufficiently aligning the reasoning flow in prompts with the complex reasoning graphs inherent in multi-step questions. 2) *Reducing reasoning difficulty*. In standard CoT without residuals, a reasoning step must a) first implicitly identify the necessary prerequisites and b) then perform reasoning on them. This dual burden can be quite demanding. In contrast, by explicitly linking necessary prerequisites using residual connections, RESPROMPT reduces the workload of a reasoning step to the core reasoning process itself, thus simplifying the mission of each step. This concept can also be analogized to human intelligence in solving multi-step questions: when provided with corresponding conditions, solving a single reasoning step becomes much easier.

## 3 EXPERIMENTS

### 3.1 EXPERIMENTAL SETUP

**Datasets.** We evaluate RESPROMPT on six benchmarks, covering three type of reasoning tasks: 1) Mathematical reasoning, including GSM8K (Cobbe et al., 2021), AQUA-RAT (Ling et al., 2017), MathQA (Amini et al., 2019), SVAMP (Patel et al., 2021); 2) Sequential reasoning, SCONE-Alchemy (Long et al., 2016); and 3) Commonsense reasoning: StrategyQA (Geva et al., 2021). Since RESPROMPT is a prompting based reasoning approach, we evaluate it on the test set of each dataset. The detailed statistics of these datasets are provided in Appendix C.1.

**Language Models.** We evaluate RESPROMPT using the LLaMA family of models, including LLaMA (Touvron et al., 2023a) and LLaMA2 (Touvron et al., 2023b). The LLaMA family is fully open-sourced, facilitating cost-effective and reproducible evaluations. Unlike OpenAI's GPT series of APIs, which undergo frequent updates and deprecation, using LLaMA ensures that the community can consistently reproduce our results. Furthermore, to contextualize our findings within the landscape of LLMs, we also present results obtained with other LLMs reported in previous studies,

Table 1: Reasoning accuracy comparison between RESPROMPT and baseline approaches. The first four rows show results from previous works. Note that since they apply CoT to different and larger LLMs, their results are not directly comparable, but we include them for reference. Numbers marked with '†' are from (Wei et al., 2022b), while numbers marked with '‡' are from (Fu et al., 2023b). A '-' symbol indicates "not applicable". Unlike other experiments on GSM8K, for LLaMA-65B with RESPROMPT (marked with '∗'), the number of few-shot exemplars is 5 instead of 8, as 8-shot exceeds the limitation of LLaMA-65B's input length. The best results for each dataset are highlighted in **boldface**, the second-best results are underlined. Relative gains are shown in green.

| | | #Params | GSM8K (8-Shot) | AQUA-RAT (4-Shot) | MathQA (4-Shot) | SCONE (2-Shot) |
|---|---|---|---|---|---|---|
| LaMDA (Thoppilan et al., 2022) | | 137B | 17.1† | 20.6† | - | - |
| GPT-3 (Brown et al., 2020) | | 175B | 55.4‡ | - | 36.0‡ | - |
| Codex (Chen et al., 2021) | | 175B | 66.6‡ | 45.3† | 47.3‡ | - |
| PaLM (Chowdhery et al., 2022) | | 540B | 58.1† | 35.8† | - | - |
| LLaMA | Standard | 65B | 13.7 | 20.8 | 24.1 | 2.8 |
| | Short CoT | 65B | 52.2 | 35.4 | 32.0 | - |
| | Long CoT | 65B | 47.1 | 33.5 | 33.0 | 13.1 |
| | RESPROMPT | 65B | **58.4 (+11.8%)**∗ | **42.5 (+20.0%)** | **34.1 (+3.3%)** | **15.1 (+15.2%)** |
| LLaMA2 | Standard | 70B | 17.4 | 31.4 | 23.2 | 5.0 |
| | Short CoT | 70B | 57.3 | 41.3 | 38.5 | - |
| | Long CoT | 70B | 52.7 | 38.1 | 38.1 | 23.3 |
| | RESPROMPT | 70B | **65.3(+13.9%)** | **44.4 (+7.5%)** | **39.2 (+1.8%)** | **24.3 (+4.3%)** |

if they are available on our used datasets. These models include LaMDA (Thoppilan et al., 2022), GPT-3 (Brown et al., 2020), Codex (Chen et al., 2021) and PaLM (Chowdhery et al., 2022).

**Prompts.** RESPROMPT is designed to incorporate residual connections in prompts for multi-step reasoning. The original CoT prompts from Wei et al. (2022b), however, cater mostly to short-step questions (1-3 steps), making it unnecessary to build residual connections. Therefore, we select questions from the training sets of benchmarks, covering a range number of reasoning steps, to design prompts for RESPROMPT. To ensure a fair comparison and validate that our improvements stem from residual connections but not simply from varied exemplar questions, we also derive CoT prompts from our selected questions. We refer to the original CoT prompts with short-step examples as "**Original CoT**", and CoT prompts with our newly selected examples as "**Derived CoT**" (it contains various number of reasoning steps). There are two special cases worth noting: 1) For MathQA, Fu et al. (2023b) designed prompts with both short and long steps, so our prompts for Original CoT and Derived CoT are both directly from (Fu et al., 2023b). 2) To the best of our knowledge, SCONE-Alchemy has not been previously studied with LLMs. Therefore, we only compare RESPROMPT with our derived CoT prompts. Details of all prompts are provided in Appendix F.

To Reviewer n8nq

## 3.2 MAIN RESULTS

**Overall Results:** RESPROMPT **significantly enhances accuracy in complex reasoning.** We compare RESPROMPT against several baseline prompting strategies, including standard prompting (without CoT rationales), Original CoT (CoT prompts with short-step examples), and Derived CoT (CoT prompts with exemplars containing various reasoning steps). The results of this comparison are detailed in Table 1. Notably, with residual connections, RESPROMPT consistently outperforms CoT based prompting, regardless of the number of reasoning steps within the CoT exemplars. Specifically, RESPROMPT achieves an average relative gain of 12.5% on LLaMA-65B and 6.8% on LLaMA2-70B across the four benchmark tests. These clear gains underscore the effectiveness of RESPROMPT in enhancing the reasoning ability of LLMs. It is note to worth that the improvements of RESPROMPT over Derived CoT validates that the improvements of RESPROMPT stem from residual connections rather than solely from using different data examples to design prompts.

To Reviewer n8nq

**Breakdown on Number of Steps:** RESPROMPT **has strong capability to perform multi-step reasoning.** RESPROMPT is intentionally proposed to improve reasoning for questions involving multiple steps. To assess RESPROMPT's performance across questions with varying complexity, we break down questions based on the number of reasoning steps into five groups: {1, 2, 3, 4, ≥5}. In Figure 3, we present both the data percentage distribution for each group and RESPROMPT's reasoning accuracy within these groups using LLaMA2-70B across the three math benchmarks (Note

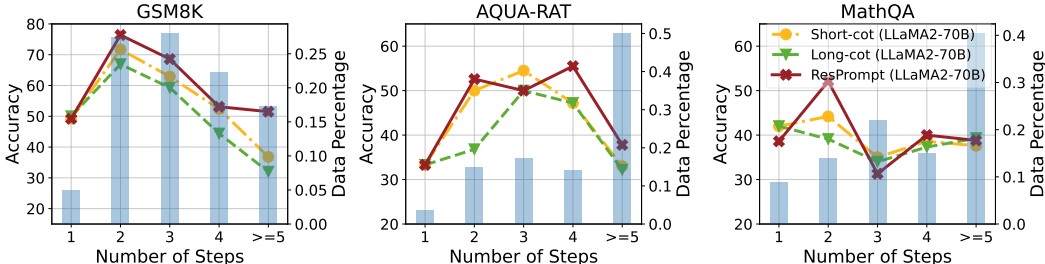

Figure 3: RESPROMPT's performance according to number of reasoning steps on GSM8K, AQUA-RAT and MathQA, all using the LLaMA2-70B model. The curves show the comparison of RESPROMPT's reasoning accuracy with CoT based baseline approaches in each step, while the blue bars represent the percentage distribution of data within each reasoning step.

that all questions in the SCONE dataset have five steps, therefore, a breakdown analysis is not necessary). We find that RESPROMPT consistently outperforms the baseline approaches in most groups. Notably, as the number of reasoning steps increases, all approaches generally experience a decline in accuracy. However, RESPROMPT demonstrates a relatively smooth decline and generally maintains higher accuracy than CoT-based approaches. In particular, for questions with $\geq 5$ reasoning steps, RESPROMPT surpasses the best CoT based approaches by achieving a remarkable improvement of 14.3% on LLaMA2-70B. This trend is similarly observed in RESPROMPT's performance on LLaMA-65B (with 21.1% gain for questions with $\geq 5$ reasoning steps), as illustrated in Appendix D.2. These results confirms RESPROMPT's strong ability for multi-step reasoning.

### 3.3 ABLATION STUDIES: HOW DOES RESPROMPT WORK?

**Where is it critical to build residual connections?** For multi-step reasoning, it might seem intuitive to include residual connections for every reasoning step. However, we aim to identify where residual connections are most critical. To investigate this, we investigate five scenarios for residual connections: 1) "*No Residual*": No residual connections (standard CoT prompting); 2) "*First Half*": Residual connections only for the first half of steps; 3) "*Second Half*": Residual connections only for the second half of steps; 4) "*Uniform*": Residual connections evenly distributed (i.e., every other step); 5) "*Full*": Residual connections in all steps. In Table 2, we report the reasoning

Table 2: Reasoning accuracy over various positions to build residual connections within RESPROMPT prompts. Results on GSM8K and AQUA-RAT are shown.

| Positions | GSM8K | | AQUA-RAT | |
|---|---|---|---|---|
| | 65B | 70B | 65B | 70B |
| No Residual | 47.1 | 52.7 | 33.5 | 38.1 |
| First Half | 54.5 | 62.7 | 31.8 | 35.0 |
| Second Half | 55.4 | 64.5 | 34.6 | 42.5 |
| Uniform | **58.4** | **65.4** | 35.8 | 38.5 |
| Full | **58.4** | 65.3 | **42.5** | **44.4** |

accuracy of these designs on GSM8K and AQUA-RAT datasets. The results reveal two key findings: 1) Building residual connections that cover the entire reasoning process is critical for achieving the highest multi-step reasoning accuracy. 2) Residual connections in later stages ("Second Half") are more important than those in early stages ("First Half"). This is reasonable since later-stage reasoning steps typically depend more on the intermediate results from earlier steps.

**How do we implement residual connections?** The implementation of residual connections plays a crucial role in fully releasing the power of RESPROMPT. In RESPROMPT, we opt to directly reuse the exact same tokens to refer to a previously mentioned intermediate result. A natural alternative approach is to use symbolic variables, namely denoting an intermediate result as 'X' and referring to it as 'X' later. In Figure 4, we compare these two implementations. The results consistently demonstrate that reusing the exact same tokens outperforms using symbolic variables on both GSM8K and AQUA-RAT benchmarks, for both LLaMA-65 and LLaMA-70 models. The worse performance of symbolic variables might be because it increases difficulty in reasoning. Understanding symbolic notation is known to be more challenging than processing semantics (Tang et al., 2023).

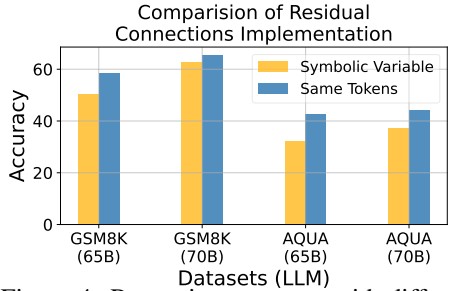

Figure 4: Reasoning accuracy with different residual connections implementations.

**How does scaling LLMs affect** RESPROMPT**?** The reasoning ability of LLMs is recognized as an "emergent ability" (Wei et al., 2022a), meaning that this capability becomes effective only when the model has sufficiently large number of parameters. In Figure 5, we explore how RESPROMPT responds to various sizes of LLaMA models, including 7B, 13B, 30B, and 65B. We derive two key observations: 1) Scaling enhances reasoning: larger model sizes consistently bring stronger reasoning performance, which echos the "emergent ability" concept. 2) RESPROMPT's advantage generally grows with size. Notably, RESPROMPT demonstrates more clear gains over CoT when applied to larger LLaMA models, particularly in the case of 65B. In contrast, with smaller LLaMA models, such as 13B and 30B on AQUA-RAT, and 30B on MathQA, RESPROMPT's performance is even worse than CoT. This indicates that the comprehension of residual connections might be part of the "emergent ability", which complements the reasoning capabilities of LLMs. Experiments with LLaMA2 yield similar results, as detailed in Appendix D.3.

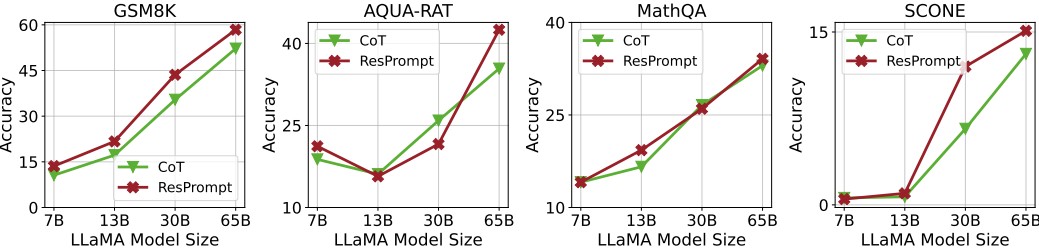

Figure 5: Reasoning accuracy comparison between RESPROMPT and CoT across all LLaMA model sizes. CoT is the model with better performance between Short CoT and Long CoT for each dataset.

**How does the number of few-shot exemplars (N-shot) affect** RESPROMPT**?** In the previous results, we maintain a fixed number of few-shot exemplars. To study the relationship between reasoning accuracy and the number of exemplars, we vary the exemplar numbers (N={2, 4, 6, 8} for GSM8K, N={1, 2, 3, 4} for AQUA-RAT and MathQA, and N={1, 2} for SCONE-Alchemy). The corresponding reasoning accuracy on LLaMA2-70B is presented in Figure 6 (results on LLaMA-65B are presented in Appendix D.4). These results demonstrate that RESPROMPT's reasoning accuracy remains mostly stable across different few-shot exemplar numbers (with exception of AQUA-RAT). Interestingly, we observe that increasing the number of few-shot exemplars can even lead to a decrease in RESPROMPT's performance (GSM8K and MathQA). This discovery implies the significance of exemplar selection, particularly the impact of various combinations of exemplars on LLM's reasoning accuracy. We leave further exploration of this area as future work.

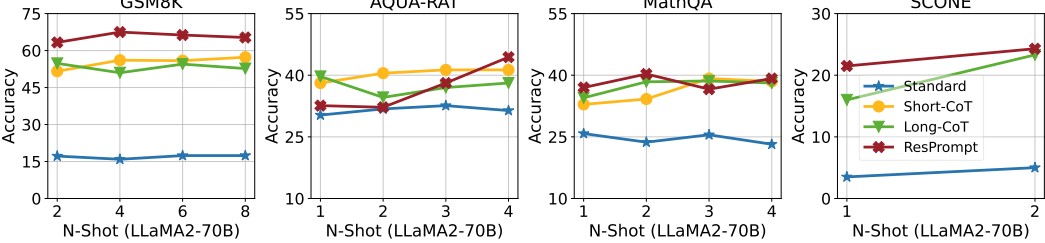

Figure 6: LLaMA2-70B's performance based on number of few-shot exemplars in RESPROMPT.

### 3.4 ANALYSIS

**Is** RESPROMPT **robust to exemplar order?** Few-shot learning in LLMs is known to be influenced by the order of exemplars (Zhao et al., 2021). Following Wei et al. (2022b), we investigate the impact of exemplar orders on RESPROMPT. We design four exemplar orders based on their number of reasoning steps: 1) "*Ascending*": Exemplars are ordered from fewer to more reasoning steps; 2) "*Descending*": Exemplars are ordered from more to fewer reasoning steps; 3) "*Alternating*": Exemplar ordering involves alternating between the least and most reasoning steps; 4) "*Ran-*

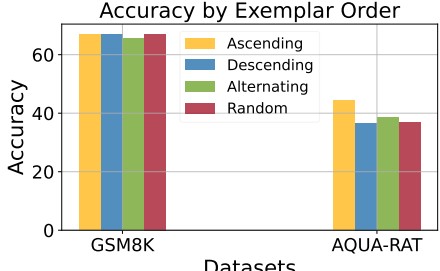

Figure 7: LLaMA2-70B reasoning accuracy with varied exemplar orders.

*dom*": Exemplars are arranged in random order. The results presented in Figure 7 demonstrate that RESPROMPT exhibits robustness to exemplar order variations in the GSM8K dataset. However, in AQUA-RAT, RESPROMPT shows slight sensitivity, with exemplars in ascending order outperforming other perturbations. This sensitivity aligns with the findings of (Zhao et al., 2021), which may be caused by LLMs' bias towards the exemplars at the end of the prompts.

---

Question: A new program had 60 downloads in the first month. The number of downloads in the second month was three times as many as the downloads in the first month, but then reduced by 30% in the third month. How many downloads did the program have total over the three months?

Correct Answer: 366.

**(a) A Multi-Step Question from GSM8K Test Set**

---

1. The program had 60 downloads in the first month. In the second month, there were 3 times as many downloads as the first month. So 3 x 60 = 180.

2. In the third month, there were 30% less downloads. 30% of 180 is 180 x 30% = 54.

3. So 180 - 54 = 126.

So the answer is 126. ❌   **(b) LLaMA2-70B Output with Chain-of-Thoughts Prompting**

---

1. In the first month, the program had 60 downloads. In the second month, the number of downloads was three times as many as the downloads in the first month, so the number of downloads was 3 * 60 = 180.

2. In the third month, the number of downloads was reduced by 30%. To find the number of downloads in the third month, we multiply the number of downloads in the second month by 0.7 (because 30% is the same as 0.3 when expressed as a decimal): 180 * 0.7 = 126.

3. So, the total number of downloads over the three months was 60 (first month) + 180 (second month) + 126 (third month) = 366. Therefore, the program had 366 downloads total over the three months.

So the answer is 366. ✅   **(c) LLaMA2-70B Output with ResPrompt Prompting (Ours)**

---

Figure 8: Case study. **(a)** A multi-step math question from GSM8K testing set. **(b)** The output of LLaMA2-70B with CoT prompts. **(c)** The output of LLaMA2-70B with RESPROMPT prompts. Connections built by LLMs with RESPROMPT are marked with colored and backgrounds .

**Case Study: Can RESPROMPT facilitate residual connection building?** In Figure 8, we present a case study using an example from the GSM8K test set to examine whether LLaMA2-70B can learn to construct residual connections, thereby enhancing multi-step reasoning capabilities. Our observations reveal that, with RESPROMPT's prompts, LLaMA2-70B successfully build residual connections in later-stage reasoning steps, leading to the correct final result. However, LLMs prompted with CoT appear to "lose direction" after the third step. We infer that this discrepancy arises from CoT's linearly structured reasoning flow. In later stages, LLMs may struggle to correctly utilize intermediate results from earlier steps, which highlights the significance of building residual connections for effective multi-step reasoning. More case studies on each dataset can be found in Appendix E.

**Error Analysis: Understanding RESPROMPT's mistakes.** In Table 3, we summarize the error types made by LLaMA2-70B with RESPROMPT on GSM8K and AQUA-RAT datasets. We analyze the first 15 wrong examples from both datasets and categorize the errors into three types: 1) "*Wrong Problem Solving*": Any types of mistakes related to the

Table 3: Statistics of Error Types in LLaMA2-70B.

| Error Type | GSM8K | AQUA-RAT |
|---|---|---|
| Wrong Problem Solving | | |
| - Wrong Reasoning Flow | 11 (73.3%) | 5 (33.3%) |
| - Build Wrong Residual Connection | 1 (6.6%) | 0 (0%) |
| - Wrong Calculation/Derivation | 1 (6.6%) | 3 (20.0%) |
| Repetition | 2 (13.3%) | 5 (33.3%) |
| Wrong Groundtruth | 0 (0%) | 2 (13.3%) |

reasoning process, including errors in reasoning flow, wrong residual connection construction, or minor calculation/derivation errors; 2) "*Repetition*": When LLMs fail to terminate and produces nonsensical contents; 3) "*Wrong Ground-truth*": The ground-truth answers contain noise. We find that the majority of errors stem from problem-solving, suggesting significant room for further enhancing the reasoning process. Repetition also accounts for a non-trivial portion, particularly in AQUA-RAT. This could be due to the relatively long and complex prompts in RESPROMPT. LLMs learn to generate longer sequences of tokens during reasoning, which may increase the risk of repetition. We provide error examples on each dataset made by LLaMA2-70B in Appendix E.

## 3.5 WHEN IS RESPROMPT NOT ESSENTIAL?

The previous results have demonstrated that RESPROMPT significantly enhances reasoning abilities, particularly for multi-step questions with complex reasoning structures. We are also interested in how RESPROMPT performs in scenarios where questions are relatively simple, or the underlying reasoning process is straightforward. To this end, we apply RESPROMPT to the SVAMP and Strat-

egyQA datasets. SVAMP is a math benchmark with questions having a maximum of two reasoning steps. StrategyQA primarily consists of multi-step commonsense reasoning questions, but their underlying reasoning flows are generally straightforward and not as complex as the graph structures in the four datasets presented in Table 1. We show an example from StrategyQA, along with their corresponding underlying reasoning flows in Appendix D.5. The reasoning accuracy of RESPROMPT and baseline approaches on these datasets is summarized in Table 4. We observe that RESPROMPT's performance is mixed in comparison to the baselines. This suggests that building residual connections within prompts may not be necessary when questions are simple or do not have complex reasoning graphs, as standard CoT is sufficient to capture their straightforward reasoning flows.

## 4 RELATED WORK

We discuss three categories of related work: "In-Context Learning and Emergent Ability", "Prompting-Based Reasoning", and "Multi-Step Reasoning". Due to space limitations, we provide a concise overview here and direct readers to Appendix B for a comprehensive review.

**In-Context Learning and Emergent Ability.** Our work focuses on more structured prompting strategy, which is closely related to in-context learning (Brown et al., 2020). It refers to LLMs' capacity to adapt from a few exemplars without model

Table 4: Comparison between RESPROMPT and baseline approaches on SVAMP and StrategyQA datasets. The best results for each dataset are highlighted in **boldface**, the second-best results are underlined. Relative gains are highlighted in green, and relative losses are marked in red.

|  | Prompting | #Params | SVAMP (8-Shot) | StrategyQA (6-Shot) |
|---|---|---|---|---|
| LLaMA | Standard | 65B | 61.4 | 70.5 |
|  | Short CoT | 65B | 68.7 | 70.0 |
|  | Long CoT | 65B | 63.2 | **71.2** |
|  | RESPROMPT | 65B | **71.1**(+3.4%) | 70.2(-1.4%) |
| LLaMA2 | Standard | 70B | 62.1 | 72.8 |
|  | Short CoT | 70B | **73.7** | **76.1** |
|  | Long CoT | 70B | 70.0 | 72.6 |
|  | RESPROMPT | 70B | 71.1(-1.4%) | 73.1(-3.9%) |

parameter changes. As models grow and train on more data, they exhibit significantly amplified performance accorss many tasks (Kaplan et al., 2020; Rae et al., 2021; Hoffmann et al., 2022; Chowdhery et al., 2022), or even obtain entirely new capabilities such as reasoning over complex questions. This phenomenon is recently termed "emergent ability" (Wei et al., 2022a).

**Prompting-Based Reasoning.** LLMs, when guided with suitable prompts, display competitive reasoning skills without requiring fine-tuning (Wei et al., 2022b; Fu et al., 2023a; Ni et al., 2023). A milestone is the CoT prompting approach (Wei et al., 2022b), which offers step-by-step rationales. While numerous enhancements have been proposed for CoT (Wang et al., 2023b; Kojima et al., 2022; Zhang et al., 2023; Gao et al., 2023; Zhou et al., 2023b), it often falls short with complex multi-step reasoning tasks (Fu et al., 2023b; Zhou et al., 2023a). Our contribution introduces a residual connection based prompting strategy, outperforming standard CoT for multi-step reasoning.

**Multi-Step Reasoning.** Simple CoT prompting struggles with complex, multi-step problems in LLMs. While Zhou et al. (2023a) and Khot et al. (2023) address this by decomposing questions and Fu et al. (2023b) integrate more complex reasoning steps and employ a majority voting mechanism, these methods generally add extra stages to reasoning. Our approach simplifies this by incorporating residual connections into prompts, facilitating a more efficient one-pass decoding process.

## 5 CONCLUSION

In this paper, we propose RESPROMPT, a new prompting strategy aimed at enhancing multi-step reasoning in LLMs. The core idea behind RESPROMPT is to reconstruct the complex reasoning graphs inherent in multi-step questions within prompts. To achieve this, we introduce "residual connection", which involves adding missing links to transform the linearly structured CoT prompts into graph-like structures to capture the complex reasoning process in multi-step questions. These residual connections are implemented by reusing the exact same tokens when referring to intermediate results from earlier steps. Our extensive experiments demonstrate that RESPROMPT significantly advances multi-step reasoning accuracy on LLaMA family of models (LLaMA and LLaMA2). We discuss the limitations of this work and potential future directions in Appendix A.

## ETHICS STATEMENT

Our work does not introduce additional ethical risks beyond those inherent in existing prompting based reasoning research. Nevertheless, as our approach is within the scope of LLMs, there remains a potential for LLMs to generate unexpected reasoning outputs. We anticipate further advancements in the field to address this concern in the future.

## REPRODUCIBILITY STATEMENT

We conduct our experiments using the LLaMA family of models, which are fully open-sourced under licenses. Additionally, all six benchmarks used in this paper are publicly available. Our study is purely based on prompting, and we have provided the prompts used for each benchmark in Table 23 to Table 28. All our experiments are using "greedy decoding" during LLMs generation. With these resources, reproducing our experiments should pose no barrier.

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

# Appendix

## Table of Contents

## A    LIMITATIONS AND FUTURE WORK

While our experiments primarily focus on the open-sourced LLaMA family of models, it is important to acknowledge that the impact of RESPROMPT on closed-sourced larger LLMs, such as GPT-4 (OpenAI, 2023) and PaLM (Chowdhery et al., 2022; Anil et al., 2023), is not clear. We hope that our work serves as a catalyst for future research endeavors in this direction. Investigating how to effectively optimize and adapt RESPROMPT for these more extensive models can pave the way for even greater breakthroughs in multi-step reasoning tasks.

## B    FULL RELATED WORK

**In-Context Learning and Emergent Ability.** Our work centers on enhancing the interdependence within prompts for complex multi-step reasoning, which is closely related to *in-context learning* (Brown et al., 2020). In-context learning describes the ability of language models to learn from a few demonstration examples and solve new tasks without the need to update the model parameters. Recent work has shown that as these models scale to larger sizes and are trained on more tokens, they exhibit stronger and even entirely new capabilities, such as reasoning over complex questions (Kaplan et al., 2020; Rae et al., 2021; Hoffmann et al., 2022; Chowdhery et al., 2022). This phenomenon is often referred to as *emergent ability* (Wei et al., 2022a). In light of this, our primary contribution lies in the effective integration of residual connections within prompts, which proves to be pivotal in addressing problems that involve multiple reasoning steps.

**Prompting-Based Reasoning.** Recent progress demonstrates that when provided with appropriate prompts, LLMs can attain competitive reasoning abilities compared to earlier approaches that rely on fine-tuning (Wei et al., 2022b; Lewkowycz et al., 2022; Fu et al., 2023a; Ni et al., 2023). A milestone in this field is chain-of-thought (CoT) prompting (Wei et al., 2022b), wherein not only the final answer but also intermediate reasoning rationales for solving a complex problem are provided in the demonstration. CoT prompting has been further improved from various angles, including implementing a majority vote mechanism across multiple sampled reasoning paths (Wang et al., 2023b), simplifying intermediate reasoning rationale into a straightforward "Let's think step by step" prompt (Kojima et al., 2022), selecting representative CoT demonstrations from each question cluster (Zhang et al., 2023), executing the reasoning steps by generating codes (Gao et al., 2023), and progressively updating the demonstration set (Zhou et al., 2023b). However, empirical findings suggest that simple CoT is less effective in solving problems that involve multi-step reasoning (Fu et al., 2023b; Zhou et al., 2023a; Khot et al., 2023). Recent work has also expanded upon CoT by organizing and processing thoughts using more complex structures, such as trees (Yao et al., 2023a; Long, 2023) and graphs (Besta et al., 2023; Yao et al., 2023b). Tree of thought (ToT) and graph of though (GoT) are more relevant for tasks that require strategic reasoning, such as backtracking, traversal, sorting, etc. The demo applications in (Yao et al., 2023a; Besta et al., 2023) include examples like sorting, document merging, game of 24, etc. On the other hand, RESPROMPT aims to capture the complex underlying structure in standard multi-step problems. Therefore, although both RESPROMPT and ToT/GoT are related to the complex "structure", RESPROMPT targets different purposes compared to ToT and GoT. We position our work within the domain of prompting-based reasoning, and propose a simple yet novel prompting strategy based on residual connections, which leads to significant improvements over CoT for multi-step reasoning.

**Multi-Step Reasoning.** LLMs have shown limitations in solving problems that require multiple steps (e.g., ≥ 5 steps in GSM8K (Cobbe et al., 2021) as in (Zhou et al., 2023a)) when using simple CoT prompting (Fu et al., 2023a;b). In response, Zhou et al. (2023a) and Khot et al. (2023) initially decompose a complex question into several sub-tasks and then address each sub-question sequentially. As an alternative approach, Fu et al. (2023b) introduce questions with higher reasoning complexity, as measured by the number of reasoning steps, into CoT prompts. They then utilize a majority voting mechanism on the most complex reasoning paths among the sampled ones to arrive at a final answer. Both approaches rely on an extra strategy beyond the intermediate reasoning steps of CoT, namely decomposition in Zhou et al. (2023a) and Khot et al. (2023) and majority voting in Fu et al. (2023b), leading to a two-stage reasoning process. In contrast, our work shows that multi-step reasoning can be significantly enhanced by incorporating appropriate residual connections just in the intermediate reasoning steps, enabling a more efficient one-pass decoding process.

## C  DETAILED EXPERIMENTAL SETTINGS

### C.1  DATASETS DETAILS.

We use six benchmarks for three type of tasks to evaluate the reasoning capability of RESPROMPT:
1) Mathematical reasoning, including GSM8K (Cobbe et al., 2021), AQUA-RAT (Ling et al., 2017),
MathQA (Amini et al., 2019), SVAMP (Patel et al., 2021); 2) Sequential reasoning, SCONE-
Alchemy (Long et al., 2016); and 3) Commonsense reasoning: StrategyQA (Geva et al., 2021).
In Table 5, we present their statistics. GSM8K, MathQA, SVAMP, SCONE-Alchemy, and Strat-
egyQA have annotations that allow us to easily compute the number of reasoning steps in each
question. For AQUA-RAT, we use annotations from (Ribeiro et al., 2023). In addition, the original
SCONE-Alchemy dataset lacks language descriptions of object states in each step, so we incorporate
the language annotations from (Ribeiro et al., 2023) to describe the intermediate results.

Table 5: Dataset statistics. Due to the large volume of MathQA (†), we randomly sample 1000 ex-
amples to accelerate evaluation. Similarly, for StrategyQA (‡), we randomly sample 800 examples.

| Dataset | Number of Samples | Number of Steps | | | | |
|---|---|---|---|---|---|---|
| | | 1-step | 2-step | 3-step | 4-step | ≥ 5-step |
| GSM8K | 1319 | 6.3% | 27.1% | 27.6% | 22.0% | 17.0% |
| AQUA-RAT | 254 | 3.5% | 15.0% | 17.3% | 14.1% | 50.0% |
| MathQA | 2985† | 8.5% | 15.2% | 21.4% | 14.4% | 40.5% |
| SVAMP | 1000 | 23.7% | 76.2% | - | - | - |
| SCONE-Alchemy | 899 | - | - | - | - | 100% |
| StrategyQA | 2289‡ | 0.8% | 27.3% | 53.2% | 15.0% | 3.7% |

### C.2  HARDWARE RESOURCES

RESPROMPT is a prompting based reasoning approach, and we only need to perform inference with
LLMs. Therefore, a single experiment of RESPROMPT on the largest model used in this paper
(LLaMA-65B and LLaMA2-70B) can be done on one *AWS p4de.24xlarge* instance with appropriate
choice of batch size (we fix the batch size to 3 for all benchmarks in this paper).

## D  EXTRA EXPERIMENTS

### D.1  REASONING ACCURACY ON LLaMA2-CHAT MODEL

Table 6: Reasoning accuracy of LLaMA2-Chat-70B on GSM8K, AQUA-RAT, MathQA and
SCONE-Alchemy datasets. The best results of LLaMA2-Chat-70B for each dataset are highlighted
in **boldface**, the second-best results are underlined. Relative gains are highlighted in green, and
relative losses are marked in red. Results of LLaMA2-70B base model are listed for reference.

| | | #Params | GSM8K | AQUA-RAT | MathQA | SCONE |
|---|---|---|---|---|---|---|
| LLaMA2 | RESPROMPT | 70B | **65.3** | **44.4** | **39.2** | **24.3** |
| LLaMA2-Chat | Standard | 70B | 13.3 | 24.4 | 24.9 | 2.2 |
| | Short CoT | 70B | 52.2 | **33.0** | 34.4 | - |
| | Long CoT | 70B | 51.8 | 32.6 | 36.1 | 11.6 |
| | RESPROMPT | 70B | **61.1(+17.0%)** | 30.7(-6.9%) | **39.6(+9.6%)** | **16.3(+40.5%)** |

In Table 6, we also provide the reasoning accuracy of LLaMA2-Chat-70B. LLaMA2-Chat-70B is
fine-tuned based on the LLaMA2-70B base model for chatbot applications. We observe a non-
trivial decline in reasoning accuracy when compared to the base model. We speculate this is be-
cause LLaMA2-Chat-70B is fine-tuned for non-reasoning purposes, and thus affect its reasoning
capability. One possible implication is that the evaluation of reasoning capabilities should ideally be
conducted within the base model or with models fine-tuned specifically for reasoning tasks.

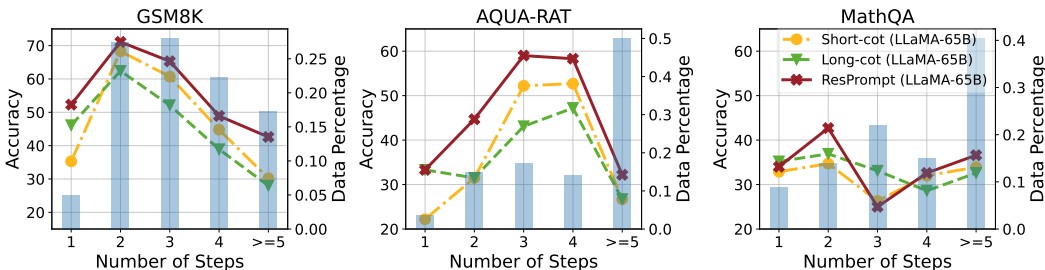

Figure 9: RESPROMPT's performance according to number of reasoning steps on GSM8K, AQUA-RAT and MathQA, all using the LLaMA-65B model. The curves show the comparison of RESPROMPT's reasoning accuracy with CoT based baseline approaches in each step, while the blue bars represent the percentage distribution of data within each reasoning step.

## D.2 ACCURACY BREAKDOWN BASED ON NUMBER OF STEPS WITH LLaMA-65B.

Figure 9 presents a breakdown of LLaMA-65B's reasoning accuracy based on the number of reasoning steps in each question. Similar to the results observed in LLaMA2-70B (as discussed in Section 3.2), RESPROMPT consistently outperforms CoT-based baselines in improving LLaMA-65B's reasoning accuracy. Notably, as the number of steps in questions increases, RESPROMPT exhibits a smoother accuracy decline compared to the baseline approaches.

## D.3 ACCURACY FOR DIFFERENT LLaMA2 SIZES

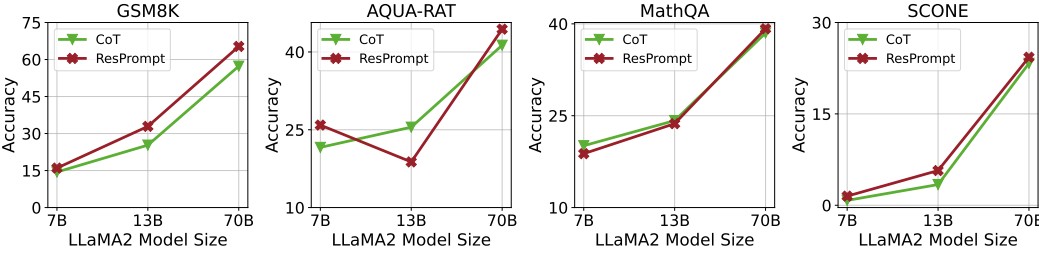

Figure 10: Reasoning accuracy comparison between RESPROMPT and CoT across all LLaMA2 models. CoT represents the better performance between Short CoT and Long CoT for each dataset.

Figure 10 illustrates how reasoning accuracy of RESPROMPT and CoT based baselines is affected by LLaMA2 model scale. Similar to the results obtained with LLaMA-65B in Section 3.3, larger models yield better overall reasoning performance. Furthermore, we consider building and understanding residual connections as an "emergent ability", following the reasoning capabilities of LLMs. This is highlighted by the observation that RESPROMPT's advantage over baselines becomes more pronounced as the model size increases, particularly at 70B. We also note that the gains on MathQA and SCONE-Alchemy datasets are not significant as they are on LLaMA-65B in Section 3.3.

## D.4 FEW-SHOT EXEMPLARS' IMPACT ON REASONING ACCURACY USING LLaMA-65B

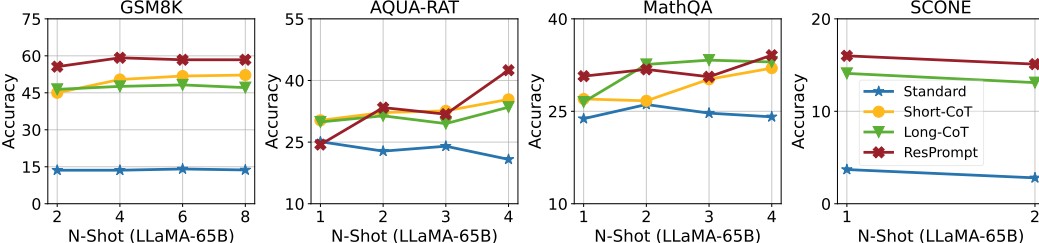

Figure 11: LLaMA-65B's performance based on number of few-shot exemplars in RESPROMPT.

In Figure 11, we compare the reasoning accuracy of RESPROMPT and CoT based approaches using the LLaMA-65B model. Similar to the results obtained with LLaMA2-70B in Section 3.3, we observe that the number of few-shot exemplars does not significantly impact reasoning accuracy. It's

worth noting that increasing the number of exemplars can sometimes lead to a decrease in LLaMA-65B's performance. This observation implies further investigation into how different combinations of exemplars affect the performance of LLMs in future studies. Note that for the GSM8K dataset, we report LLaMA-65B's 5-shot accuracy for the 6-shot and 8-shot positions in Figure 11. This adjustment is necessary because RESPROMPT's prompts with more than 5 exemplars exceed the token length limitation of LLaMA-65B (2048).

## D.5 AN EXAMPLE AND ITS REASONING FLOW FROM THE STRATEGYQA DATASET

In Figure 12, we present a multi-step common-sense reasoning example from the StrategyQA dataset, along with its corresponding underlying reasoning flow. Despite having multiple reasoning steps, the question's underlying reasoning flow is nearly linear. This observation may help explain why RESPROMPT does not provide improvements on StrategyQA dataset, as standard CoT is sufficient to reconstruct the nearly linear underlying reasoning flow.

## D.6 COMPARISON TO FINE-TUNED LLAMA

It is also interesting to compare the reasoning capability of RESPROMPT with fine-tuned based approaches. Since our experiments are conducted on LLaMA family of models, we compare RESPROMPT to TÜLU (Wang et al., 2023c). TÜLU is a fine-tuned model based on LLaMA (v1) and spans various parameter scales (7B, 13B, 30B, and 65B).

The results comparison on GSM8K dataset are shown in Table 7 (GSM8K is the only common dataset shared by this work with TÜLU (Wang et al., 2023c)). We notice that fine-tuned TÜLU still outperforms RESPROMPT. However, this performance gap significantly narrows when using the 65B model. This observation echos our earlier findings in Section 3.3 and Appendix D.3, indicating RESPROMPT's ability to construct and understand residual connections appears to be an "emergent ability".

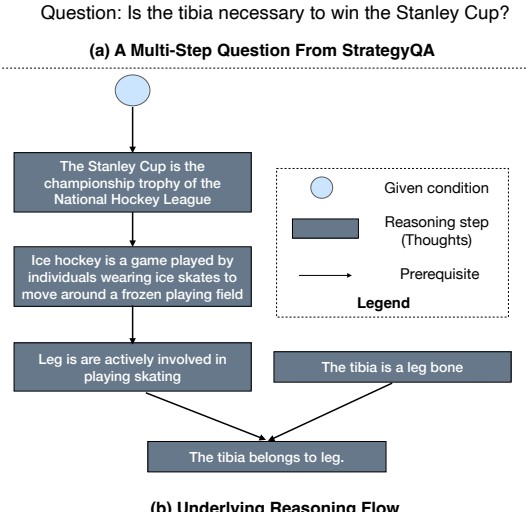

Question: Is the tibia necessary to win the Stanley Cup?

**(a) A Multi-Step Question From StrategyQA**

**(b) Underlying Reasoning Flow**

Figure 12: StrategyQA data example. **(a)** A multi-step question. **(b)** Its underlying reasoning flow.

Table 7: Performance on GSM8K compared with TÜLU, a fine-tuned model based on LLaMA. TÜLU is prompted with 8-shot CoT. The numbers marked with † are from (Wang et al., 2023c).

|  | 7B | 13B | 30B | 65B |
|---|---|---|---|---|
| TÜLU-CoT | 27.0† | 36.5† | 51.0† | 60.0† |
| LLaMA |  |  |  |  |
| -CoT | 10.9 | 20.1 | 37.1 | 52.2 |
| -RESPROMPT | 13.6 | 21.7 | 43.0 | 58.4 |

## D.7 HOW DOES NOISE IN PROMPTS AFFECT RESPROMPT?

Most LLM prompts are human-crafted, leading to inevitable noise from annotation errors. We explore the impact of noise on RESPROMPT by introducing two perturbations into prompts: 1) Incorrect numbers in reasoning steps, and 2) Linking prerequisites in later stages to incorrect early results. As in Table 8, RESPROMPT proves robust to noise in GSM8K, echoing findings from (Min et al., 2022; Wang et al., 2023a;

Table 8: RESPROMPT performance under noise in prompts on GSM8K and AQUA-RAT datasets.

| Prompts | GSM8K | | AQUA-RAT | |
|---|---|---|---|---|
|  | **65B** | **70B** | **65B** | **70B** |
| RESPROMPT |  |  |  |  |
| -w/ noise | 56.1 | 64.4 | 28.3 | 36.6 |
| -w/o noise | 58.4 | 65.3 | 42.5 | 44.4 |

Madaan & Yazdanbakhsh, 2022) that prompt format often outweighs intermediate result accuracy. However, a noticeable accuracy dip is seen in AQUA-RAT, hinting at dataset-dependent noise sensitivity. A more comprehensive investigation of this phenomenon is left for future research.

Table 9: Comparison between RESPROMPT and complexity based prompting on GSM8K dataset. 8-step represents all exemplars in the prompts are questions requiring 8 reasoning steps, while 8&9-step stands for a mix of 8-step and 9-step examples in prompts, and 9-step means all exemplars are 9-reasoning step questions. All prompts for complexity based prompting are from the official repository `https://github.com/FranxYao/chain-of-thought-hub`

| | #Params | Complexity 8-step (8-Shot) | Complexity 8&9-step (8-Shot) | RESPROMPT (8-Shot) | Complexity 9-step (4-Shot) | RESPROMPT (4-Shot) |
|---|---|---|---|---|---|---|
| LLaMA | 65B | 48.3 | 49.6 | **58.4** | 54.5 | **59.2** |
| LLaMA2 | 70B | 64.2 | 63.8 | **65.3** | 63.5 | **67.5** |

## D.8  MORE EXPERIMENTS ON GSM8K.

**Compare to complexity based prompting (Fu et al., 2023b).** Using more complex examples to design prompts has been shown beneficial to reasoning (Fu et al., 2023b). In Table 9, we compare RESPROMPT with three versions of complexity based prompting approach. The results demonstrate that RESPROMPT consistently outperforms all the three versions of complexity based prompting. This comparison is also an ablation study that confirms that the significant improvement of RESPROMPT over CoT stems from correctly building the residual connections rather than solely from selecting more powerful examples to design prompts.

To Reviewer n8nq, Reviewer w7DW and Reviewer z49G

Table 10: Comparison between RESPROMPT and multi-step reasoning baselines on GSM8K dataset. We directly use the prompts as originally specified in the respective papers.

| | #Params | Decomp (1-Shot) | RESPROMPT (1-Shot) | Least to Most (4-Shot) | RESPROMPT (4-Shot) |
|---|---|---|---|---|---|
| LLaMA | 65B | 40.4 | 46.6 | 53.6 | **58.4** |
| LLaMA2 | 70B | 50.3 | 57.2 | **60.1** | 67.5 |

**Compare to advanced multi-step baselines.** To understand the performance of RESPROMPT compared to approaches that use multiple stages prompting for multi-step reasoning (Khot et al., 2023; Zhou et al., 2023a), we conduct experiments on GSM8K dataset. The results, presented in Table 10, consistently demonstrate that RESPROMPT outperforms these advanced baselines for multi-step reasoning. Note that these baselines aim to decompose a complex question into several sub-questions, while RESPROMPT still maintains one pass flow via a more powerful problem solving process.

To Reviewer w7DW and Reviewer z49G

**Cost-performance analysis.** Despite the RESPROMPT's superiority in multi-step reasoning performance, it also raises concerns about the inference cost. In Table 11, we compare the relative inference cost, including number of tokens and inference speed between RESPROMPT and baselines.

Table 11: Relative comparison of inference cost on GSM8K dataset using LLaMA2-70B.

| | # Tokens | Inference Speed | Accuracy |
|---|---|---|---|
| Original-CoT | 1 | 1 | 57.3 |
| Complexity | 3.76X | 0.56X | 64.2 |
| RESPROMPT | 3.06X | 0.65X | 65.3 |

To Reviewer n8nq

On average, the number of combined tokens of prompts and outputs of RESPROMPT is about 3.06X more than the tokens in the original CoT (Wei et al., 2022b) on the entire GSM8K test set, while the inference speed of RESPROMPT is about 0.65X of original CoT. We acknowledge that our prompt is longer than the original CoT and thus has higher inference cost. However, compared to complexity based prompting (Fu et al., 2023b), RESPROMPT only has 3.06X/3.76X = 0.81X tokens and is 0.65X/0.56X = 1.16X faster in terms of inference speed, while achieving a better performance.

**Performance with self-consistency strategy.** Self-consistency (Wang et al., 2023b) has been shown to be powerful in further improving reasoning performance by reaching an agreement between several decoding paths. In Table 12, we compare RESPROMPT and CoT with self-consistency (5-path) on GSM8K dataset. The

Table 12: Performance comparison with self-consistency on GSM8K dataset.

| | #Param | CoT-SC (8-Shot) | RESPROMPT-SC (8-Shot) |
|---|---|---|---|
| LLaMA | 65B | 54.0 | 58.0 |
| LLaMA2 | 70B | 64.0 | 72.0 |

To Reviewer z49G

results show that with self-consistency can further boost the performance of RESPROMPT. In addition, with self-consistency, RESPROMPT still achieves clearly higher reasoning accuracy than CoT.

**Performance on GPT family of models.** We're also curious whether RESPROMPT still has superitoy in more capable LLMs such as OpenAI's GPT-3.5 and GPT-4 (OpenAI, 2023). We compare vanilla CoT and RESPROMPT using the "gpt-3.5-turbo-0613" and "gpt-4-0613" models on GSM8K dataset. The results, shown in Table 13, demonstrate that ResPrompt is also beneficial for the most powerful OpenAI LLMs.

Table 13: Performance on GPT LLMs on GSM8K dataset.

|  | CoT (8-Shot) | RESPROMPT (8-Shot) |
|---|---|---|
| GPT-3.5 | 73.0 | 76.0 |
| GPT-4 | 91.0 | 93.0 |

To Reviewer z49G

### D.9 ADDITIONAL EXPERIMENTS ON CSQA AND HOTPOTQA.

To Reviewer w7DW

Table 14: Comparison between RESPROMPT and baselines on CSQA and HotpotQA datasets.

|  | CSQA (Acc) (4-Shot) | HotpotQA (EM/F1) (4-Shot) |
|---|---|---|
| Standard | 78.6 | 33.3/43.2 |
| CoT | 79.0 | 37.3/43.2 |
| RESPROMPT | 78.0 | 36.3/49.9 |

To further understand RESPROMPT's ability for reasoning tasks requiring extensive knowledge, we conduct comparision between RESPROMPT and CoT on CSQA (Talmor et al., 2019) and HotpotQA (Yang et al., 2018) benchmarks. We show the results in Table 14. These results demonstrate that RESPROMPT can just achieve comparable performance to the baselines on both CSQA and HotpotQA. This observation is not surprising since both benchmarks primarily require extensive knowledge to answer the questions, rather than complex multi-step reasoning. Therefore, it is natural that RESPROMPT may not be essential in these knowledge assessment benchmarks.

# E    MORE CASE STUDIES ON EACH DATASET

## E.1    GSM8K

Table 15: Examples of correct outputs by LLaMA2-70B on GSM8K dataset.

**Question:** Judy teaches 5 dance classes, every day, on the weekdays and 8 classes on Saturday. If each class has 15 students and she charges $15.00 per student, how much money does she make in 1 week?

**Model Answer:** Judy teaches 5 classes every day on the weekdays, so she teaches 5 (classes per day) * 5 (days per week) = 25 classes on the weekdays. Each class has 15 students, so she teaches 25 (classes on the weekdays) * 15 (students per class) = 375 students on the weekdays. She charges $15 per student, so she makes 375 (students on the weekdays) * $15 (charge per student) = $5625 on the weekdays. On Saturday, she teaches 8 classes. Each class has 15 students, so she teaches 8 (classes on Saturday) * 15 (students per class) = 120 students on Saturday. She charges $15 per student, so she makes 120 (students on Saturday) * $15 (charge per student) = $1800 on Saturday. Adding these together, she makes $5625 (on the weekdays) + $1800 (on Saturday) = $7425 in a week. So, Judy makes $7425 in a week. So the answer is 7425. ✅

**Question:** In a dance class of 20 students, 20% enrolled in contemporary dance, 25% of the remaining enrolled in jazz dance, and the rest enrolled in hip-hop dance. What percentage of the entire students enrolled in hip-hop dance?

**Model Answer:** First, let's calculate how many students enrolled in contemporary dance. 20% of the 20 students enrolled in contemporary dance, so the number of students who enrolled in contemporary dance is 20 (total students) * 0.20 (20%) = 4 students. Next, let's calculate how many students enrolled in jazz dance. 25% of the remaining students enrolled in jazz dance, so the number of students who enrolled in jazz dance is 20 (total students) - 4 (students who enrolled in contemporary dance) * 0.25 (25%) = 4 students. Finally, let's calculate how many students enrolled in hip-hop dance. The remaining students enrolled in hip-hop dance, so the number of students who enrolled in hip-hop dance is 20 (total students) - 4 (students who enrolled in contemporary dance) - 4 (students who enrolled in jazz dance) = 12 students. Now, let's calculate the percentage of students who enrolled in hip-hop dance. The percentage of students who enrolled in hip-hop dance is 12 (students who enrolled in hip-hop dance) / 20 (total students) = 0.60 (60%). Therefore, 60% of the students enrolled in hip-hop dance. So the answer is 60. ✅

Table 16: Examples of wrong outputs by LLaMA2-70B on GSM8K dataset.

**Question:** Josh decides to try flipping a house. He buys a house for $80,000 and then puts in $50,000 in repairs. This increased the value of the house by 150%. How much profit did he make?

**Model Answer**: First, let's calculate the value of the house after the repairs. The value of the house increased by 150%, so the new value is 150% of the original value. To find this, we multiply the original value by 1.50 (because 150% is the same as 1.50 when expressed as a decimal): $80,000 (original value) * 1.50 = $120,000 (new value). Now, let's calculate Josh's profit. He bought the house for $80,000 and put in $50,000 in repairs, so his total investment was $80,000 + $50,000 = $130,000. His profit is the difference between the new value of the house and his total investment: $120,000 (new value) - $130,000 (total investment) = -$10,000. So, Josh made a loss of $10,000. So the answer is -10000. ❌

**Question:** John drives for 3 hours at a speed of 60 mph and then turns around because he realizes he forgot something very important at home. He tries to get home in 4 hours but spends the first 2 hours in standstill traffic. He spends the next half-hour driving at a speed of 30mph, before being able to drive the remaining time of the 4 hours going at 80 mph. How far is he from home at the end of those 4 hours?

**Model Answer:** First, let's calculate how far John drove in the first 3 hours. He drove at a speed of 60 mph, so he drove 60 (mph) * 3 (hours) = 180 miles. Then, he turned around and drove for 4 hours. He spent the first 2 hours in standstill traffic, so he didn't move at all during that time. Then, he drove at a speed of 30 mph for half an hour, so he drove 30 (mph) * 0.5 (half an hour) = 15 miles. Finally, he drove at a speed of 80 mph for the remaining time of the 4 hours, which is 4 (hours) - 0.5 (half an hour) = 3.5 hours. So, he drove 80 (mph) * 3.5 (hours) = 280 miles. Adding these together, he drove 180 (miles in the first 3 hours) + 15 (miles in the next half-hour) + 280 (miles in the remaining 3.5 hours) = 475 miles. Therefore, John is 475 miles from home at the end of those 4 hours. So the answer is 475. ❌

## E.2   AQUA-RAT

Table 17: Examples of correct outputs by LLaMA2-70B on AQUA-RAT dataset.

**Question:** Alex and Jacob works at a toy shop that make toys. Alex takes 7 hours to make a toy, and Jacob takes 9 hours to make a toy. During a month, both of them makes 35 toys in total. If both of them have worked for almost similar number of hours how many toys have been prepared by Jacob? Answer Choices: (a) 15 (b) 16 (c) 17 (d) 18 (e) 19

**Model Answer:** The total number of toys made by both of them is 35. Alex takes 7 hours to make a toy, and Jacob takes 9 hours to make a toy. So, the total number of hours that Alex and Jacob have worked is: 35 toys × (7 hours + 9 hours) = 35 toys × 16 hours = 560 hours. Since they have worked for almost similar number of hours, we can assume that Alex and Jacob have worked for 560/2 = 280 hours each. Given that Jacob takes 9 hours to make a toy, the number of toys that Jacob has made is: 280 hours ÷ 9 hours/toy = 31.11 toys. Since Jacob has made 31.11 toys, he has made 31 toys and 11% of a toy. So, the number of toys that Jacob has made is: 31 toys + 11% of a toy = 31 toys + 11% × 1 toy = 31 toys + 0.11 toy = 31.11 toys. So, the number of toys that Jacob has made is 31.11 toys. So the answer is (a). ✅

**Question:** A hollow cube of size 5cm is taken, with the thickness of 1cm. It is made of smaller cubes of size 1cm .If the outer surface of the cube is painted how many faces of the smaller cubes remain unpainted? Answer Choices: (a) 438 (b) 550 (c) 500 (d) 450 (e) 498

**Model Answer:** The hollow cube has 6 faces, and each face has 4 smaller cubes. So, the total number of smaller cubes on the faces is: 6 faces * 4 smaller cubes/face = 24 smaller cubes. Now, the hollow cube has 12 edges, and each edge has 2 smaller cubes. So, the total number of smaller cubes on the edges is: 12 edges * 2 smaller cubes/edge = 24 smaller cubes. Now, the hollow cube has 8 vertices, and each vertex has 3 smaller cubes. So, the total number of smaller cubes on the vertices is: 8 vertices * 3 smaller cubes/vertex = 24 smaller cubes. Adding up the number of smaller cubes on the faces, edges, and vertices: 24 smaller cubes (faces) + 24 smaller cubes (edges) + 24 smaller cubes (vertices) = 72 smaller cubes. Now, the hollow cube has 24 smaller cubes on the faces, edges, and vertices, and the total number of smaller cubes is 72. So, the number of smaller cubes that are not on the faces, edges, and vertices is: 72 smaller cubes - 24 smaller cubes = 48 smaller cubes. So, the answer is (a). ✅

Table 18: Examples of wrong outputs by LLaMA2-70B on AQUA-RAT dataset.

**Question:** A train running at a speed of 100 miles/hour, takes 10 hours to reach its destination. After covering quarter of the distance, it starts raining and the train has to be slowed to speed of 75 miles/hour. What is the total journey duration? Answer Choices: (a) 10 (b) 11.5 (c) 12.5 (d) 13.5 (e) 15

**Model Answer:** The train has to cover 100 miles/hour * 10 hours = 1000 miles. After covering 1/4 of the distance, the train has to cover 3/4 of the distance. The train has to cover 3/4 of the distance at 75 miles/hour. So, the time taken to cover 3/4 of the distance is: 3/4 * 1000 miles / 75 miles/hour = 300 miles / 75 miles/hour = 4 hours. So, the total journey duration is: 10 hours + 4 hours = 14 hours. So the answer is (d). ❌

**Question:** The original retail price of an appliance was 60 percent more than its wholesale cost. If the appliance was actually sold for 20 percent less than the original retail price, then it was sold for what percent more than its wholesale cost? Answer Choices: (a) 20% (b) 28% (c) 36% (d) 40% (e) 42%

**Model Answer:** The original retail price was 60% more than its wholesale cost. So, the original retail price is: 100% + 60% = 160% of the wholesale cost. Now, the appliance was actually sold for 20% less than the original retail price. So, the actual selling price is: 100% - 20% = 80% of the original retail price. Given that the original retail price is 160% of the wholesale cost, the actual selling price is: 80% of 160% = 128% of the wholesale cost. So, the appliance was sold for 128% more than its wholesale cost. So the answer is (c). ❌

### E.3 MATHQA

Table 19: Examples of correct outputs by LLaMA2-70B on MathQA dataset.

**Question:** an angry arjun carried some arrows for fighting with bheeshm . with half the arrows , he cut down the arrows thrown by bheeshm on him and with 6 other arrows he killed the chariot driver of bheeshm . with one arrow each he knocked down respectively the chariot , the flag and the bow of bheeshm . finally , with one more than 4 times the square root of arrows he laid bheeshm unconscious on an arrow bed . find the total number of arrows arjun had . Options: (a) 90 , (b) 100 , (c) 110 , (d) 120 , (e) 130 ,

**Model Answer:** Let's think step by step. Let's assume the total number of arrows Arjun had is x. Given that Arjun cut down the arrows thrown by Bheeshm on him with half the arrows. So, the number of arrows thrown by Bheeshm on him = x/2. Given that Arjun killed the chariot driver of Bheeshm with 6 other arrows. So, the number of arrows Arjun had = x + 6. Given that Arjun knocked down the chariot, the flag and the bow of Bheeshm with one arrow each. So, the number of arrows Arjun had = x + 3. Given that Arjun laid Bheeshm unconscious on an arrow bed with one more than 4 times the square root of arrows. So, the number of arrows Arjun had = $x + 1 + 4\sqrt{x}$. Now, we have 4 equations with 4 unknowns. We can solve them to get the value of x. $x + 6 = x + 3 = x + 1 + 4\sqrt{x} = x/2$, $6 = 3 = 1 + 4\sqrt{x} = x/2$, $3 = 1 + 4\sqrt{x} = x/2$, $2 = 4\sqrt{x} = x/2$, $\sqrt{x} = 1/2 = x/2$, x = 2, x = 2. So, the total number of arrows Arjun had is 2. So the answer is (b), 100. ✅

**Question:** the time taken by a man to row his boat upstream is twice the time taken by him to row the same distance downstream . if the speed of the boat in still water is 45 kmph , find the speed of the stream ? Options: (a) 12 kmph , (b) 13 kmph , (c) 14 kmph , (d) 15 kmph , (e) 16 kmph ,

**Model Answer:** Let's think step by step. Let's assume the speed of the stream is x km/hr. Then the speed of the boat upstream = 45-x km/hr and the speed of the boat downstream = 45+x km/hr. Given that the time taken by a man to row his boat upstream is twice the time taken by him to row the same distance downstream. This means the time taken to row upstream is 2 times the time taken to row downstream. Therefore, we can set up the equation: (45-x)/(45+x) = 2. Solving this equation, we get x = 15. So the speed of the stream is 15 km/hr. So the answer is (d), 15 km/hr. ✅

Table 20: Examples of wrong outputs by LLaMA2-70B on MathQA dataset.

**Question:** bag contains 7 green and 8 white balls . if two balls are drawn simultaneously , the probability that both are of the same colour is - . Options: (a) 7 / 15 , (b) 2 / 8 , (c) 7 / 11 , (d) 13 / 5 , (e) 87 ,

**Model Answer:** Let's think step by step. The probability of drawing two balls of the same color is the probability of drawing two green balls plus the probability of drawing two white balls. The probability of drawing two green balls is 7/15 (the probability of drawing the first green ball is 7/15 and the probability of drawing the second green ball is 6/14). The probability of drawing two white balls is 8/15 (the probability of drawing the first white ball is 8/15 and the probability of drawing the second white ball is 7/14). Therefore, the probability of drawing two balls of the same color is 7/15 + 8/15 = 15/15 = 1. So the answer is (e), 1. ❌

**Question:** the product of two numbers is 2028 and their h . c . f is 13 . the number of such pairs is : Options: (a) 1 , (b) 2 , (c) 3 , (d) 4 , (e) 5 ,

**Model Answer:** Let's think step by step. Let's assume the two numbers are x and y. Given that their product is 2028, we have xy = 2028. Given that their h.c.f. is 13, we have x = 13a and y = 13b, where a and b are integers. Therefore, xy = 13a * 13b = 13(ab). Since xy = 2028, we have 13(ab) = 2028. This means ab = 2028/13 = 156. Since a and b are integers, ab = 156 can have the following solutions: a = 1, b = 156; a = 2, b = 78; a = 3, b = 52; a = 4, b = 39; a = 6, b = 26; a = 9, b = 17; a = 12, b = 13. Therefore, the number of pairs of numbers is 7. So the answer is (e), 7. ❌

### E.4 SCONE-ALCHEMY

Table 21: Examples of correct outputs by LLaMA2-70B on SCONE-Alchemy dataset.

**Question:** The world contains 7 beakers. Each beaker may contain up to 4 units of colored chemical. The chemical can be poured into another beaker, drained away, or mixed. The initial state of each beaker is: First beaker has 1 orange chemical; Second beaker has 2 green chemicals; Third beaker has 0 chemicals; Fourth beaker has 2 red chemicals; Fifth beaker has 4 red chemicals; Sixth beaker has 4 green chemicals; Seventh beaker has 1 purple chemical. We perform the following actions: Remove two units from the fourth beaker.Remove three green units from the 6th beaker.Remove one unit from the second beaker.Add the first beaker to the second one.Mix the second beaker. Question: what are the chemicals in each beaker?

**Model Answer:** The initial state of each beaker is: First beaker has 1 orange chemical; Second beaker has 2 green chemicals; Third beaker has 0 chemicals; Fourth beaker has 2 red chemicals; Fifth beaker has 4 red chemicals; Sixth beaker has 4 green chemicals; Seventh beaker has 1 purple chemical. Actions: 1) Remove two units from the fourth beaker. Before this action, the fourth beaker has 2 red chemicals. After this action, the fourth beaker has 0 red chemicals. 2) Remove three green units from the 6th beaker. Before this action, the sixth beaker has 4 green chemicals. After this action, the sixth beaker has 1 green chemical. 3) Remove one unit from the second beaker. Before this action, the second beaker has 2 green chemicals. After this action, the second beaker has 1 green chemical. 4) Add the first beaker to the second one. Before this action, the first beaker has 1 orange chemical; and the second beaker has 1 green chemical. After this action, the first beaker has 0 orange chemicals; and the second beaker has 2 chemicals whose color is not clear at this moment. 5) Mix the second beaker. Previous results show that before this action, the second beaker has 2 chemicals whose color is not clear. After this action, the second beaker has 2 brown chemicals. So the answer is: First beaker has 0 orange chemicals, denoted as 1:_; second beaker has 2 brown chemicals, denoted as 2:bb; third beaker has 0 chemicals, denoted as 3:_; fourth beaker has 0 red chemicals, denoted as 4:_; fifth beaker has 4 red chemicals, denoted as 5:rrrr; sixth beaker has 1 green chemical, denoted as 6:g; seventh beaker has 1 purple chemical, denoted as 7:p. The answer of these 7 beakers can also be written in the following format: 1:_ 2:bb 3:_ 4:_ 5:rrrr 6:g 7:p.✅

Table 22: Examples of wrong outputs by LLaMA2-70B on SCONE-Alchemy dataset.

**Question:** The world contains 7 beakers. Each beaker may contain up to 4 units of colored chemical. The chemical can be poured into another beaker, drained away, or mixed. The initial state of each beaker is: First beaker has 2 yellow chemicals; Second beaker has 4 green chemicals; Third beaker has 0 chemicals; Fourth beaker has 2 purple chemicals; Fifth beaker has 1 red chemical; Sixth beaker has 4 orange chemicals; Seventh beaker has 0 chemicals. We perform the following actions: Throw out three parts of the orange chemical.Pour the red chemical into the orange chemical.Mix the orange chemical.Pour out 1 part of green chemical.Pour out two parts of green chemical. Question: what are the chemicals in each beaker?

**Model Answer:** The initial state of each beaker is: First beaker has 2 yellow chemicals; Second beaker has 4 green chemicals; Third beaker has 0 chemicals; Fourth beaker has 2 purple chemicals; Fifth beaker has 1 red chemical; Sixth beaker has 4 orange chemicals; Seventh beaker has 0 chemicals. Actions: 1) Throw out three parts of the orange chemical. Before this action, the sixth beaker has 4 orange chemicals. After this action, the sixth beaker has 1 orange chemical. 2) Pour the red chemical into the orange chemical. Before this action, the sixth beaker has 1 orange chemical; and the fifth beaker has 1 red chemical. After this action, the sixth beaker has 2 chemicals whose color is not clear at this moment; and the fifth beaker has 0 chemicals. 3) Mix the orange chemical. Previous results show that before this action, the sixth beaker has 2 chemicals whose color is not clear. After this action, the sixth beaker has 2 orange chemicals. 4) Pour out 1 part of green chemical. Before this action, the second beaker has 4 green chemicals. After this action, the second beaker has 3 green chemicals. 5) Pour out two parts of green chemical. Previous results show that before this action, the second beaker has 3 green chemicals. After this action, the second beaker has 1 green chemical. So the answer is: First beaker has 2 yellow chemicals, denoted as 1:yy; second beaker has 1 green chemical, denoted as 2:g; third beaker has 0 chemicals, denoted as 3:_; fourth beaker has 2 purple chemicals, denoted as 4:pp; fifth beaker has 0 chemicals, denoted as 5:_; sixth beaker has 2 orange chemicals, denoted as 6:oo; seventh beaker has 0 chemicals, denoted as 7:_. The answer of these 7 beakers can also be written in the following format: 1:yy 2:g 3:_ 4:pp 5:_ 6:oo 7:_. ❌

# F    FULL PROMPTS OF RESPROMPT

## F.1    GSM8K AND SVAMP

Table 23: Few-shot exemplars of RESPROMPT for GSM8K and SVAMP–Part 1

Answer the following questions.

**Question:** Tina makes $18.00 an hour. If she works more than 8 hours per shift, she is eligible for overtime, which is paid by your hourly wage + 1/2 your hourly wage. If she works 10 hours every day for 5 days, how much money does she make?

**Answer:** Tina makes $18.00 an hour for the first 8 hours of each day, and she gets overtime for anything beyond 8 hours. For 8 hours a day at $18.00 an hour, she makes: 8 hours/day * $18.00/hour = $144.00/day. For the overtime, she works an extra 2 hours a day (since she works 10 hours and the first 8 are not considered overtime). Overtime is paid at her hourly wage plus half her hourly wage, which is $18.00 + $9.00 = $27.00 per hour. So for 2 hours of overtime each day, she makes: 2 hours/day * $27.00/hour = $54.00/day. Therefore, in one day, she makes: $144.00/day + $54.00/day = $198.00/day. Over the course of 5 days, she would make: 5 days * $198.00/day = $990.00. So, Tina would make $990.00 if she worked 10 hours a day for 5 days. So the answer is 990.

**Question:** Tobias is buying a new pair of shoes that costs $95. He has been saving up his money each month for the past three months. He gets a $5 allowance a month. He also mows lawns and shovels driveways. He charges $15 to mow a lawn and $7 to shovel. After buying the shoes, he has $15 in change. If he mows 4 lawns, how many driveways did he shovel?

**Answer:** He has $15 left after buying the shoes that cost $95, which means he had $95 + $15 = $110 in total before the purchase. Next, we calculate the amount he earned from his allowance. He has been saving for three months, getting $5 each month, so from his allowance he earned 3 * $5 = $15. Then we know he mowed 4 lawns and he charges $15 per lawn, so from mowing lawns, he earned 4 * $15 = $60. Let's sum up the money he earned from his allowance and from mowing lawns: $15 (allowance) + $60 (mowing) = $75. Now, let's subtract the total amount of money earned from allowance and mowing from his total amount of money before buying the shoes to find out how much he earned from shoveling driveways: $110 (total before buying shoes) - $75 (earned from allowance and mowing) = $35. As he charges $7 to shovel a driveway, we divide the total earned from shoveling by the charge per driveway to find out the number of driveways he shoveled: $35 / $7 = 5 driveways. So the answer is 5.

**Question:** Brennan was researching his school project and had to download files from the internet to his computer to use for reference. After downloading 800 files, he deleted 70% of them because they were not helpful. He downloaded 400 more files but again realized that 3/5 of them were irrelevant. How many valuable files was he left with after deleting the unrelated files he downloaded in the second round?

**Answer:** First, let's figure out how many files Brennan kept after the first download. He initially downloaded 800 files, but he deleted 70% of them. So, the number of files he kept is 30% of the original 800. To find this, we multiply 800 by 0.30 (because 30% is the same as 0.30 when expressed as a decimal): 800 * 0.30 = 240 files. Then, Brennan downloaded 400 more files. However, he realized that 3/5 of them were not useful. So, he only kept 2/5 of the new files (because 1 - 3/5 = 2/5). To find this number, we multiply 400 by 2/5: 400 * 2/5 = 160 files. So after deleting the unrelated files he downloaded in the second round, he was left with 240 files from the first download and 160 files from the second download: 240 + 160 = 400 files. Therefore, Brennan was left with 400 valuable files after deleting all the unrelated files. So the answer is 400.

**Question:** Noah is a painter. He paints pictures and sells them at the park. He charges $60 for a large painting and $30 for a small painting. Last month he sold eight large paintings and four small paintings. If he sold twice as much this month, how much is his sales for this month?

**Answer:** First, let's calculate how much Noah earned last month. From selling large paintings, he earned 8 (number of large paintings) * $60 (price per large painting) = $480. From selling small paintings, he earned 4 (number of small paintings) * $30 (price per small painting) = $120. So, his total sales for last month were $480 (from large paintings) + $120 (from small paintings) = $600. This month, he sold twice as many paintings. So his sales this month were 2 * $600 (last month's sales) = $1200. Therefore, Noah's sales for this month is $1200. So the answer is 1200.

Table 24: Few-shot exemplars of RESPROMPT for GSM8K and SVAMP–Part 2

**Question:** Betty is saving money for a new wallet which costs $100. Betty has only half of the money she needs. Her parents decided to give her $15 for that purpose, and her grandparents twice as much as her parents. How much more money does Betty need to buy the wallet?

**Answer:** Her parents give her $15, and her grandparents give her twice that amount, which is 2 * $15 = $30. So, the total amount of money she has after receiving money from her parents and grandparents is $50 (her own money) + $15 (from parents) + $30 (from grandparents) = $95. Now, to find out how much more money she needs, we subtract the total amount of money she has from the cost of the wallet: $100 (cost of the wallet) - $95 (total amount of money she has) = $5. Therefore, Betty needs $5 more to buy the wallet. So the answer is 5.

**Question:** Julie is reading a 120-page book. Yesterday, she was able to read 12 pages and today, she read twice as many pages as yesterday. If she wants to read half of the remaining pages tomorrow, how many pages should she read?

**Answer:** Yesterday, Julie read 12 pages. Today, she read twice as many pages as she did yesterday, so she read 2 * 12 = 24 pages today. So far, she has read 12 (yesterday's pages) + 24 (today's pages) = 36 pages in total. The book is 120 pages long, so she still has 120 (total pages) - 36 (read pages) = 84 pages left to read. She wants to read half of the remaining pages tomorrow, so she should read 84 (remaining pages) / 2 = 42 pages. Therefore, Julie should read 42 pages tomorrow. So the answer is 42.

**Question:** It takes Roque two hours to walk to work and one hour to ride his bike to work. Roque walks to and from work three times a week and rides his bike to and from work twice a week. How many hours in total does he take to get to and from work a week with walking and biking?

**Answer:** First, let's calculate how much time Roque spends walking to work and back home. Each walk takes two hours and he walks both ways, so each day he spends 2 (hours per walk) * 2 (there and back) = 4 hours walking. He walks three times a week, so he spends 4 (hours per day) * 3 (days per week) = 12 hours per week walking. Next, let's calculate how much time Roque spends riding his bike to work and back home. Each bike ride takes one hour, and he rides both ways, so each day he spends 1 (hour per ride) * 2 (there and back) = 2 hours biking. He rides his bike twice a week, so he spends 2 (hours per day) * 2 (days per week) = 4 hours per week biking. Adding these together, Roque spends 12 (hours per week walking) + 4 (hours per week biking) = 16 hours getting to and from work each week. So, Roque spends 16 hours in total getting to and from work each week. So the answer is 16.

**Question:** James writes a 3-page letter to 2 different friends twice a week. How many pages does he write a year?

**Answer:** Each letter James writes is 3 pages long. He writes to 2 different friends, so that's 3 (pages per letter) * 2 (letters) = 6 pages. Since he writes these letters twice a week, in a week, he writes 6 (pages per letter writing session) * 2 (letter writing sessions per week) = 12 pages. To calculate how many pages he writes in a year, we have to multiply the number of pages he writes in a week by the number of weeks in a year. There are 52 weeks in a year, so James writes 12 (pages per week) * 52 (weeks in a year) = 624 pages in a year. So, James writes 624 pages in a year. So the answer is 624.

## F.2 AQUA-RAT

Table 25: Few-shot exemplars of RESPROMPT for AQUA-RAT

Answer the following questions.

**Question:** A person is traveling at 20 km/hr and reached his destiny in 2.5 hr then find the distance? Answer Choices: (a) 53 km (b) 55 km (c) 52 km (d) 60 km (e) 50 km

**Answer:** This person traveled at 20 km/hr for 2.5 hr, so the distance that this person traveled would have been 20 km/hr * 2.5 hrs = 50 km. So the answer is (e).

**Question:** A clock shows the time as 9 a.m. If the minute hand gains 5 minutes every hour, how many minutes will the clock gain by 5 p.m.? Answer Choices: (a) 30 min (b) 35 min (c) 45 min (d) 40 min (e) 55 min
**Answer:** The clock shows at 9 a.m., and until 5 p.m., it has been 8 hours. As the minute hand gains 5 minutes every hour, in these 8 hours, it will gain 5 min/hour * 8 hours = 40 mins. So the answer is (d).

**Question:** A sum of money at simple interest amounts to Rs. 815 in 3 years and to Rs. 854 in 4 years. The sum is: Answer Choices: (a) 600 (b) 698 (c) 675 (d) 688 (e) 900

**Answer:** The amount after 3 years is Rs. 815, and the amount after 4 years is Rs. 854. The difference in amounts between the 3rd and 4th year is the simple interest for one year, because the principal remains constant in simple interest. So, the simple interest for one year is: Rs. 854 - Rs. 815 = Rs. 39. Now, if the interest for one year is Rs. 39, then the interest for 3 years is: 3 * Rs. 39 = Rs. 117. Given that the amount after 3 years (which includes the principal and 3 years of interest) is Rs. 815, the principal (or the sum we want to find) is: Rs. 815 (amount after 3 years) - Rs. 117 (3 years of interest) = Rs. 698. Therefore, the sum is Rs. 698. So the answer is (b).

**Question:** The entrance fee for a fair is $5 for persons under the age of 18, and 20% more for persons older. Each ride at the fair costs $0.50. If Joe goes with her 6 years old twin brothers, and they each took 3 rides in total. How much money does Joe end up spending at the fair? Answer Choices: (a) 16 (b) 20.5 (c) 17.5 (d) 20 (e) 4.5

**Answer:** For entrance fee, given Joe's twin brothers are both 6 years old, they will pay this under-18 entrance fee: 2 * $5 = $10. For Joe (assuming Joe is 18 or older), the entrance fee is 20% more, which is $5 + ($5 * 0.20) = $5 + $1 = $6. Adding up the entrance fees: $6 (Joe) + $10 (twins) = $16. For ride cost, each of them took 3 rides and each ride needed $0.5: 3 rides × $0.50/ride = $1.50. Since there are three of them (Joe + 2 brothers): 3 * $1.50 = $4.50 in total for all rides. Now, adding up the entrance fee and ride cost: $16 (entrance fee) + $4.50 (ride cost) = $20.50. So, Joe ends up spending $20.50. So the answer is (b).

### F.3 MathQA

Table 26: Few-shot exemplars of RESPROMPT for MathQA

Answer the following questions.

**Question:** there were 35 students in a hostel . due to the admission of 7 new students the expenses of the mess were increased by rs . 84 per day while the average expenditure per head diminished by re 1 . what was the original expenditure of the mess ? Options: (a) rs 450 , (b) rs 920 , (c) rs 550 , (d) rs . 630 , (e) none of these

**Answer:** Let's think step by step. let the original average expenditure be x rupees. Given there were 35 students originally, the total expenditure for the mess was 35x rupees. After 7 new students were admitted, the number of students became 35 + 7 = 42. According to the information, the average expenditure per head then diminished by re 1. This means the new average expenditure is x - 1 rupees per student. Therefore, the new total expenditure for the mess with 42 students is 42(x - 1) rupees. It's also given that due to the admission of 7 new students, the expenses of the mess increased by rs 84 per day. So we have 42(x - 1) = 35x + 84 7 x = 126, x = 18. So, the original average expenditure per student was rs 18. Thus, the original expenditure of the mess was: 35 students × rs 18/student = rs 630. So the answer is (d), rs . 630.

**Question:** a train 200 m long passes a man , running at 5 km / hr in the same direction in which the train is going , in 10 seconds . the speed of the train is ? Options: (a) 28 , (b) 50 , (c) 77 , (d) 22 , (e) 12

**Answer:** Let's think step by step. The train takes 10 seconds to pass the man. When a train passes an object, it covers a distance equal to its own length relative to that object. Therefore, in 10 seconds, the train covers a distance of 200m (its own length) relative to the man. So the speed of the train relative to man is ( 200 / 10 ) m/s = 20 m/s. To convert this speed from m/s to km/hr, we multiply by 18/5. So, the relative speed in km/hr = 20*(18/5) km/hr = 72 km/hr. The relative speed is the difference between the train's speed and the man's speed because they are moving in the same direction. Let's assume the speed of the train is x km/hr. Thus, the relative speed = x-5 km/hr. Since we already know the relative speed is 72 km/hr, we can have 72 km/hr = x-5 km/hr km/hr, x = 77 km / hr. So, the speed of the train is 77 km/hr. So the answer is (c), 77.

**Question:** solution x contains 20 % of material a and 80 % of material b . solution y contains 30 % of material a and 70 % of material b . a mixture of both these solutions contains 22 % of material a in the final product . how much solution x is present in the mixture ? Options: (a) 40 % , (b) 60 % , (c) 80 % , (d) 100 % , (e) 110 %

**Answer:** Answer: Let's think step by step. we can assume the total weight of the mixture = 100. Then let's denote the weight of solution x is w and the weight of solution y as 100-w (since the total weight of the mixture is 100). From the problem, solution x has 20% of Material A, which means 0.20w of Material A. And solution y has 30% of Material A, which means 0.30(100-w) of Material A. The mixture has 22% of Material A. This means that the mixture has 22% * 100 (total weight of the mixture) = 22 units of Material A. Therefore, using the above information, we can set up the equation: 0.20w + 0.30(100-w) = 22 -0.1 w = -8, w = 80. since we assumed the weight of the mixture = 100. Therefore presence of a in the mixture = 80/100 = 80%. So the answer is (c), 80%.

**Question:** a trader sells 40 metres of cloth for rs . 8200 at a profit of rs . 35 per metre of cloth . how much profit will the trder earn on 40 metres of cloth ? Options: (a) rs . 950 , (b) rs . 1500 , (c) rs . 1000 , (d) rs . 1400 , (e) none of these

**Answer:** Let's think step by step. Given that profit per metre of cloth = Rs. 35. For 40 metres of cloth, the profit will be: Profit = 40 metres x Rs. 35/metre = Rs. 1400. Thus, the trader will earn a profit of Rs. 1400 on 40 metres of cloth. So the answer is (d) Rs. 1400.

## F.4 SCONE-ALCHEMY

Table 27: Few-shot exemplars of RESPROMPT for SCONE-Alchemy

Answer the following questions.

**Question:** The world contains 7 beakers. Each beaker may contain up to 4 units of colored chemical. The chemical can be poured into another beaker, drained away, or mixed. The initial state of each beaker is: First beaker has 3 green chemicals; Second beaker has 0 chemicals; Third beaker has 0 chemicals; Fourth beaker has 0 chemicals; Fifth beaker has 1 orange chemical; Sixth beaker has 3 orange chemicals; Seventh beaker has 4 green chemicals. We perform the following actions: Throw out two units of first beaker. Throw out fifth beaker. Throw out first one. Throw out orange beaker. Throw out one unit of green. Question: what are the chemicals in each beaker?

**Answer:** The initial state of each beaker is: First beaker has 3 green chemicals; Second beaker has 0 chemicals; Third beaker has 0 chemicals; Fourth beaker has 0 chemicals; Fifth beaker has 1 orange chemical; Sixth beaker has 3 orange chemicals; Seventh beaker has 4 green chemicals. Actions: 1) Throw out two units of the first beaker. Before this action, the first beaker has 3 green chemicals, after this action, the first beaker has 1 green chemical. 2) Throw out the fifth beaker. Before this action, the fifth beaker has 1 orange chemical. After this action, the fifth beaker has 0 chemicals. 3) Throw out the first one. Previous results show that before this action, the first beaker has 1 green chemical. After this action, the first beaker has 0 green chemicals. 4) Throw out the orange beaker. Before this action, the sixth beaker has 3 orange chemicals. After this action, the sixth beaker has 0 chemicals. 5) Throw out one unit of green. Previous knowledge shows that currently only the seventh beaker has 4 green chemicals. After this action, the seventh beaker has 3 green chemicals. So the answer is: First beaker has 0 chemicals, denoted as 1:_; second beaker has 0 chemicals, denoted as 2:_; third beaker has 0 chemicals denoted as 3:_; fourth beaker has 0 chemicals, denoted as 4:_; fifth beaker has 0 chemicals, _; sixth beaker has 0 chemicals, denoted as 6:_; seventh beaker has 3 green chemicals, denoted as 7:ggg. The answer of these 7 beakers can also be denoted as the following format: 1:_ 2:_ 3:_ 4:_ 5:_ 6:_ 7:ggg.

**Question:** The world contains 7 beakers. Each beaker may contain up to 4 units of colored chemical. The chemical can be poured into another beaker, drained away, or mixed. The initial state of each beaker is: First beaker has 2 yellow chemicals; Second beaker has 3 orange chemicals; Third beaker has 4 purple chemicals; Fourth beaker has 1 orange chemical; Fifth beaker has 4 green chemicals; Sixth beaker has 3 red chemicals; Seventh beaker has 3 purple chemicals. We perform the following actions: Throw out one unit of last beaker. Throw out three units of third one. Pour fourth beaker into it. It turns brown. Throw out one unit of it. Question: what are the chemicals in each beaker?

**Answer:** The initial state of each beaker is: First beaker has 2 yellow chemicals; Second beaker has 3 orange chemicals; Third beaker has 4 purple chemicals; Fourth beaker has 1 orange chemical; Fifth beaker has 4 green chemicals; Sixth beaker has 3 red chemicals; Seventh beaker has 3 purple chemicals. Actions: 1) Throw out one unit of the last beaker. Before this action, the seventh beaker has 3 purple chemicals. After this action, the seventh beaker has 2 purple chemicals. 2) Throw out three units of the third one. Before this action, the third beaker has 4 purple chemicals. After this action, the third beaker has 1 purple chemical. 3) Pour fourth beaker into it. Before this action, the third beaker has 1 purple chemical; and the fourth beaker has 1 orange chemical. After this action, the third beaker has 2 chemicals whose color is not clear at this moment; and the fourth beaker has 0 chemicals. 4) It turns brown. Previous results show that before this action, the third beaker has 2 chemicals whose color is not clear. After this action, the third beaker has 2 brown chemicals. 5) Throw out one unit of it. Previous results show that before this action, the third beaker has 2 brown chemicals. After this action, the third beaker has 1 brown chemical. So the answer is: First beaker has 2 yellow chemicals, denoted as 1:yy; second beaker has 3 orange chemicals, denoted as 2:ooo; third beaker has 1 brown chemical, denoted as 3:b; fourth beaker has 0 chemicals, denoted as 4:_; fifth beaker has 4 green chemicals, denoted as 5:gggg; sixth beaker has 3 red chemicals, denoted as 6:rrr; seventh beaker has 2 purple chemicals, denoted as 7:pp. The answer of these 7 beakers can also be written in the following format: 1:yy 2:ooo 3:b 4:_ 5:gggg 6:rrr 7:pp.

## F.5 STRATEGYQA

Table 28: Few-shot exemplars of RESPROMPT for StrategyQA

Answer the following questions.

**Question:** Will the Albany in Georgia reach a hundred thousand occupants before the one in New York?

**Answer:** Albany, Georgia had a population hovering around 75,000, and population had been gradually declining. Albany, New York, had a population around 100,000. Albany, New York's population had also been experiencing a gradual decline. Since Albany, New York is closer to the 100,000 number than Albany, Georgia, and they both have population decline, it seems unlikely that Albany, Georgia would reach 100,000 occupants before Albany, New York. So the answer is no.

**Question:** Would a student of the class of 2017 have amnesia about 9/11?

**Answer:** The event of 9/11 occurred on September 11, 2001. Class of 2017 high school students are round the age of 17 or 18 in 2017, so they would have been born in 1999 or 2000. Students born in 1999 or 2000 would have been 1 or 2 years old in 2001, the year of event 9/11. Children typically have amnesia before the age of 3 or 4. Since Class of 2017 students were 1 or 2 years old, which is younger than the age of 3 or 4, they would have amnesia about event 9/11. So the answer is yes.

**Question:** Would baker's dozen of side by side Mac Trucks jam up Golden Gate Bridge?

**Answer:** A baker's dozen is traditionally 13. The width of a Mac truck is around 8 feet. The width of the Golden Gate Bridge is around 90 feet. If 13 Mack Trucks were placed side by side, they would occupy a width of about 13*8=104 feet. Since 104 feet (13 Mack Trucks placed side by side) is wider than 90 feet (Golden Gate Bridge), it would cause traffic jam. So the answer is yes.

**Question:** Is the tibia necessary to win the Stanley Cup?

**Answer:** The Stanley Cup is awarded to the team that wins the NHL playoffs. To win NHL playoffs, a team needs effective players who can skate. Effective skating generally requires two functional legs. The tibia is a crucial bone in the leg. Since healthy tibia is eseential for functional legs, tibia is necessary for skating. So the answer is yes.

**Question:** Could a dandelion suffer from hepatitis?

**Answer:** Hepatitis is an inflammation of the liver, typically affect mammals. Dandelions are flowering plants. Since flowering plants and mammals are very different biologically, Dandelions can not have hepatitis. So the answer is no..

**Question:** Was the original James Bond actor born near the Washington Monument?

**Answer:** The original actor to portray James Bond in the official film series was Sean Connery. Sean Connery was born in Fountainbridge, Edinburgh, Scotland. The Washington Monument is located in Washington, D.C., United States. Edinburgh, Scotland, and Washington, D.C., United States, are thousands of miles apart and are in two separate countries. Since Sean Connery was born in Edinburgh and the Washington Monument is in Washington, D.C., it's clear that the original James Bond actor was not born near the Washington Monument. So the answer is no.

