# OpenReview forum: "ResPrompt: Residual Connection Prompting Advances Multi-Step Reasoning in Large Language Models"
_ICLR.cc/2024/Conference — ICLR 2024 Conference Withdrawn Submission_

### Official Review · Reviewer_z49G · 2023-10-30

**Soundness:** 2 fair
**Presentation:** 3 good
**Contribution:** 2 fair
**Rating:** 3
**Confidence:** 3

**Summary:**

This work aims to improve the multi-step reasoning ability of Chain-of-Thought (CoT) prompting. When coping with complex problems, CoT tends to only leverage the reasoning outcomes from one prior steps and ignores those from several prior steps. Thus, CoT only exploits the linear reasoning structure in problem-solving and struggles to look back on several steps earlier, whose reasoning scheme is represented as a graph. To address this issue, this work introduces a new prompting method, called RESPrompt, to explicitly add connections to previous reasoning steps in prompting exemplars. Experiments on the family of Llama models show the performance improvement of RESPrompt over the Vallina CoT method.

**Strengths:**

Enhancing the complex reasoning ability of CoT-like prompting methods is a crucial problem in prompt-based reasoning of Large Language Models (LLMs). It is promising to construct the graph-like reasoning scheme and back-track prior reasoning steps to facilitate next-step inference for complex tasks.

**Weaknesses:**

However, the reviewer has several concerns about the Methodology and Experiment.
Methodology:
1. First, the Vallina CoT seems to already leverage the reasoning results from previous steps for next-step inference, as shown in Figure 2 (b). In Step 4, the Vallina CoT uses the results obtained in Step 1,2,3 to infer the money Tobias earned from shoveling driveways. Thus, the reviewer is quite skeptical that Vallina CoT can intrinsically leverage the results from several prior steps for next-step inference, especially on GPT-3.5 and GPT-4 models with increased model capability.
2. Concerns about Novelty and Generality. Although RESPrompt is claimed to match the graph structure of reasoning in complex tasks, it also revokes a linearly step-by-step reasoning scheme using more crafted CoT-like prompting examples. Thus, the reviewer is concerned with the novelty of RESPrompt. Moreover, it may require extensive labor and expertise to 1. manually design the prompt examples for RESPrompt and 2. establish optimal residual connections within multi-step reasoning. These issues make RESPrompt less generalizable to new tasks.
3. What are the 'provided question conditions' in Line 11, Page 4?


Experiments:
1. RESPrompt is only compared with the Vallina CoT method. Empirical comparison between RESPrompt with representative methods such as Least2Most, and CoT-SC is lacking.
2. Moreover, the reviewer is curious about whether the performance gain of RESPrompt over the Vallina CoT diminishes when testing on more powerful models such as GPT 3.5 and GPT 4.
3. The performance gain of RESPrompt over the Vallina CoT is marginal. As shown in Figure 6, RESPrompt only enjoys significant performance gains in specific few-shot prompting settings on some datasets. This raises concerns about the effectiveness of RESPrompt in general settings and tasks.

**Questions:**

The authors are encouraged to address the concerns in Weaknesses.

---

> ### Author Response · Authors · 2023-11-22
> **Response to Reviewer z49G - Part1**
>
> Dear Reviewer z49G,
>
> We sincerely appreciate your in-depth and insightful feedback and comments. In the following, we provide new experiments and explanations to your concerns one-by-one. We really appreciate your suggestions and hope our response could address your concerns. We also welcome further discussion if you have any additional questions.
>
>
> ### **Methodology:**
>
> > **W1: Vanilla  CoT seems to already leverage the reasoning results from previous steps in Figure 2 (b). Vallina CoT can intrinsically leverage the results from several prior steps especially on GPT-3.5 and GPT-4 models with increased model capability.**
>
>
> **A1:** We greatly appreciate the reviewer for this valuable feedback. We would like to clarify that Figure 2 (b) **is one CoT Prompt, not the output of LLMs**. We showcase this CoT prompt in Figure 2 (b) to illustrate that its reasoning flow is linear (Figure 2 (c)), and therefore cannot recover the reasoning graph shown in Figure 2 (d). Instead, we present the output of LLMs with vanilla CoT prompts in Figure 8. In the step 3 of Figure 8, the LLaMA model fails to understand the meaning of the results from previous steps when using CoT prompts alone. In contrast, with ResPrompt, the model is able to establish correct connections with the previous results in step 3 and derive the correct final outcome. This case study in Figure 8 demonstrates that vanilla CoT prompting fails to effectively utilize the results from prior steps.
>
>
> To assess whether ResPrompt continues to enhance multi-step reasoning compared to vanilla CoT on more capable LLMs, we conduct comparison experiments between ResPrompt and vanilla CoT using GPT3.5 and GPT4 on the GSM8K dataset. The results, presented in the following table, demonstrate that ResPrompt is effective in improving complex reasoning even for more capable OpenAI models.
>
>
> | | Vanilla CoT (8-shot)  | ResPrompt (8-shot) |
> |:-----------:|:----------:|:------------------------:|
> |GPT3.5| 73.0 | 76.0 |
> |GPT4|  91.0 | 93.0|
>
> Since the reviewer also asks a question for this experiment in the following, we will present more details of this experiment in the following.
>
>
>
>
> > **W2: Novelty concern: ResPrompt also revokes a linearly step-by-step reasoning scheme and Generality concern: ResPrompt requires extensive labor and expertise.**
>
>
> **A2:** We sincerely thank the reviewer for this in-depth and valuable comment.
>
> For novelty, we would like to clarify using the prompt example in Figure 2. When comparing the reasoning flow of CoT (Figure 2 (c)) with the underlying reasoning graph of the question (Figure 2 (d)), it shows that the linear structure in Figure 2 (c) misses several links necessary to reconstruct the graph structure in Figure 2 (d). ResPrompt is designed to add these missing links. It is important to note that although ResPrompt builds upon the linear prompt of CoT, it enables the model to realize the necessary intermediate results in earlier steps by repeating them. This repeating operation effectively adds the missing links and helps the model become aware of the graph structure. In summary, the novelty of ResPrompt lies in its **ability to recover the underlying complex reasoning graph of multi-step questions, which CoT is unable to achieve**.
>
>
> For generality, we would like to express our appreciation to the reviewer for raising this question. We would like to clarify that while ResPrompt indeed requires labor and expertise to design the prompts, this is **not a unique limitation of our approach**. To the best of our knowledge, almost all prompting-based reasoning approaches in LLMs require extensive labor and expertise [1-4]. For example, in the vanilla CoT paper [1], the authors manually design prompts for each benchmark. And in [4], the authors manually work principles to select examples to build prompts. However, we agree that the automatic generation of ideal prompts for reasoning is a promising research question for future study.
>
>
> > **W3: What are the 'provided question conditions' in Line 11, Page 4?.**
>
>
> **A3:** We really thank the reviewer for seeking clarification on this point. "Provided question conditions" refers to the conditions explicitly given in the question itself, rather than intermediate results derived from earlier steps during the problem-solving process. For example, in step 6 of Figure 2 (e), two conditions are required: 1) "```$35``` earned from shoveling" and 2) "```$7``` to shovel". Here the ```$35``` “earned from shoveling” is an intermediate result from earlier steps, while “he charges ```$7``` to shovel” is a given condition from the question in Figure 2(a).

---

> > ### Author Response · Authors · 2023-11-22
> > **Response to Reviewer z49G - Part2**
> >
> > ### **Experiments:**
> >
> > > **W4: Compare to representative methods such as Least2Most, and CoT-SC.**
> >
> > **A4:** We sincerely appreciate the reviewer for providing this insightful suggestion. We agree that including these comparisons will enhance the comprehensiveness and robustness of our experiments. We have conducted a comparison on GSM8K between our proposed ResPrompt and the Least to Most prompting [2] and Self-Consistency [3] approaches. The results of these comparisons are presented in the following tables.
> > .
> >
> > - Compared to Least to Most prompting [2]:
> > | | Least_to_most (4-shot) | ResPrompt (4-shot) |
> > |:-----------:|:----------:|:------------------------:|
> > |LLaMA-65B| 53.6       |   59.2                |
> > |LLaMA2-70B| 60.1       |  67.5  |
> >
> >
> >
> >
> > - Compared to Self-Consistency [3]. Please note that self-consistency is not a prompting strategy, but an improvement in the decoding stage, which can be integrated into any prompting approach. Therefore, we compare the performance of both CoT and ResPrompt with self-consistency, i,e., CoT-SC and ResPrompt-SC.
> > | | CoT-SC (8-shot)  | ResPrompt-SC (8-shot) |
> > |:-----------:|:----------:|:------------------------:|
> > |LLaMA-65B| 54.0 | 58.0 |
> > |LLaMA2-70B|  64.0 | 72.0|
> >
> > The results in the above two tables demonstrate that ResPrompt outperforms advanced prompting baselines (ResPrompt vs Least to Most ) and CoT in the self-consistency setting (ResPrompt-SC vs CoT-SC). Due to resource limitations during this rebuttal period, we randomly selected 100 examples from GSM8K for the self-consistency experiments. We will add the experiments on full benchmark in the revised paper.
> >
> >
> > > **W5: Experiments on more powerful models such as GPT 3.5 and GPT 4.**
> >
> > **A5:** We greatly appreciate the reviewer's valuable suggestion. We compare Vanilla CoT and ResPrompt using the ```gpt-3.5-turbo-0613``` and ```gpt-4-0613``` models on GSM8K. The results, shown in the following table, demonstrate that ResPrompt is also beneficial for the most powerful OpenAI LLMs. Due to the slow budget administration constraints during the limited rebuttal period, the author had to use their own resources for this experiment. As a result, we only randomly selected 100 examples from GSM8K for quick experiments. After the budget administration issue is addressed later, we will include experiments on the full benchmark in the revised paper.
> >
> >
> > | | Vanilla CoT (8-shot)  | ResPrompt (8-shot) |
> > |:-----------:|:----------:|:------------------------:|
> > |GPT3.5| 73.0 | 76.0 |
> > |GPT4|  91.0 | 93.0|

---

> > > ### Author Response · Authors · 2023-11-22
> > > **Response to Reviewer z49G - Part3**
> > >
> > > > **W6: ResPrompt only enjoys significant performance gains in specific few-shot prompting settings on some datasets.**
> > >
> > > **A6:** We greatly thank the reviewer for providing this in-depth comment. As shown in Figure 6, ResPrompt indeed may not always outperform baseline approaches, particularly on the AQUA-RAT dataset. However, we would like to clarify that the setting in Table 1 follows the conventional approach used in previous LLM reasoning work [1] for performance comparison, where 4-shot is used for AQUA-RAT.
> > >
> > > In addition, we would like to explain the instability of AQUA-RAT when using a small number of exemplars.
> > >
> > > We think there are two possible reasons.
> > >
> > > -  Unlike GSM8K, where most questions follow a similar math word format, AQUA-RAT has a more diverse type of question. It includes standard math word problems as well as more formal questions, such as "If m > 0, y > 0, and x is m percent of 4y, then, in terms of y, m is what percentage of x? A)y/400, B)4y, C)50y, D)2500/y, E)5000/y". Consequently, it becomes more challenging for the few-shot exemplars to cover all the diverse question types in AQUA-RAT. For example, if the 1-shot exemplar is a pure math word problem, LLMs may adhere to the math word problem format and are more likely to make mistakes when encountering the more formal question type mentioned above. This is particularly relevant to ResPrompt because its problem-solving process is more complex than that of CoT. LLMs may become stick to on the intricate reasoning process and, as a result, are less likely to solve other question types.
> > >
> > > - The test set of AQUA-RAT only has 254 examples. Compared to other benchmarks that mostly have 1000+ examples, the performance of AQUA-RAT may be more sensitive to different few-shot settings, especially for ResPrompt where the reasoning process in exemplars is more complex.
> > >
> > >
> > >
> > >
> > >
> > > \
> > > \
> > > [1] Jason Wei, Xuezhi Wang, Dale Schuurmans, Maarten Bosma, Brian Ichter, Fei Xia, Ed H. Chi, Quoc V. Le, and Denny Zhou. Chain-of-thought prompting elicits reasoning in large language models. In NeurIPS, 2022b.
> > > [2] Denny Zhou, Nathanael Schärli, Le Hou, Jason Wei, Nathan Scales, Xuezhi Wang, Dale Schuurmans, Claire Cui, Olivier Bousquet, Quoc V. Le, and Ed H. Chi. Least-to-most prompting enables complex reasoning in large language models. In  ICLR 2023,
> > > [3] Xuezhi Wang, Jason Wei, Dale Schuurmans, Quoc Le, Ed Chi, Sharan Narang, Aakanksha Chowdhery, Denny Zhou. Self-consistency improves chain of thought reasoning in language models ICLR 2023.
> > > [4] Yao Fu, Hao Peng, Ashish Sabharwal, Peter Clark, and Tushar Khot. Complexity-based prompting for multi-step reasoning. In ICLR 2023

---

### Official Review · Reviewer_BjpG · 2023-10-30

**Soundness:** 4 excellent
**Presentation:** 4 excellent
**Contribution:** 4 excellent
**Rating:** 8
**Confidence:** 3

**Summary:**

This paper proposes a residual connection prompting method for enhancing multi-step reasoning in large language models. The proposed ResPrompt aims at reconstructing the reasoning graph within prompts, which is implemented by adding missing links to transform the linearly structured CoT prompts into graph-like structures. Experiments are conducted via LLaMA and LLaMA2 on 6 benchmarks covering mathematical reasoning, sequential reasoning, and commonsense reasoning. The experimental results show the benefits of the proposed ResPrompt and some interesting findings.

**Strengths:**

The proposed approach uses a simple method to recover complex reasoning graphs and reduce reasoning difficulty, which is easy to implement. According to the experimental results, the proposed approach shows significant improvements. The ablation study and analysis provide interesting findings.

**Weaknesses:**

Please see the questions listed below.

**Questions:**

Q1: For the performances according to the number of reasoning steps (Figure 3), it is shown that ResPrompt performances are fluctuant as the number of reasoning steps grows in the MathQA dataset, and in the questions with steps >= 5 it shows similar performance with the compared approaches. What are the possible reasons for this observation?

---

> ### Author Response · Authors · 2023-11-22
> **Response to Reviewer BjpG**
>
> Dear Reviewer BjpG,
>
> We sincerely appreciate your supportive and positive feedback on our paper! Please find our response in the following .We hope this explanation could further address your comments.
>
> > **W1: ResPrompt performances are fluctuant as the number of reasoning steps grows in the MathQA.**
>
> **A1:** We sincerely appreciate the reviewer for this comment. We think the fluctuation in MathQA can be explained in the following two perspectives:
> - According to [1], the annotation in MathQA dataset is much noisier, so we use the examples in [1] to build Long-CoT and Short-CoT to avoid unintended noise for MathQA dataset. However, these examples from [1] are already after a careful example selection based on complexity, and therefore is already a very strong approach. We believe achieving better results with ResPrompt compared to this baseline actually further shows the effectiveness of ResPrompt.
>
> - The results are also model-dependent. Figure 3 shows the results on LLaMA2-70B, while results on LLaMA-65B are shown in Figure 9 in appendix. We notice that ResPrompt has a more clear improvement on >=5 step questions on LLaMA-65B. Therefore, one specific result of one dataset may be model-dependent, but generally, ResPrompt is able to improve the multi-step reasoning performance.
>
> \
> \
> [1] Fu et al, Complexity-Based Prompting for Multi-Step Reasoning, ICLR 2023.

---

### Official Review · Reviewer_w7DW · 2023-10-30

**Soundness:** 3 good
**Presentation:** 3 good
**Contribution:** 2 fair
**Rating:** 6
**Confidence:** 3

**Summary:**

This paper introduces RESPROMPT, a new prompting strategy that enhances multi-step reasoning in large language models (LLMs). By incorporating residual connections into prompts, RESPROMPT captures the complex reasoning graphs inherent in multi-step problems. Experimental results demonstrate significant improvements in reasoning accuracy, particularly for complex multi-step tasks, which confirms its motivation.

**Strengths:**

1.The proposed method is simple yet effective. By using the same tokens in intermediate steps to represent the connections between edges in the graph logic structure, it enhances the model's ability to solve complex multi-step reasoning problems.

2.The experimental results partially demonstrate the effectiveness of the method, but it could be more persuasive to include a broader range of baselines and benchmarks to further support the findings (see Weakness).

**Weaknesses:**

1.The paper's baselines only include conventional CoT methods (no matter standard, long, or short), lacking comparisons with some other approaches specifically designed to enhance CoT multi-step reasoning ability (e.g. [1][2][3]). This limitation weakens the solidity of the experimental results. Additionally, while the last paragraph in Introduction briefly mentions the advantages of the proposed method over approaches that introduce tree or graph structures during the reasoning stage, the experiments lack a proper comparison and analysis in this regard.

[1] Least-to-Most Prompting Enables Complex Reasoning in Large Language Models, ICLR 2023

[2] Complexity-Based Prompting for Multi-Step Reasoning, ICLR 2023

[3] Decomposed Prompting: A Modular Approach for Solving Complex Tasks, ICLR 2023

2.The proposed method in this paper appears to be effective primarily for mathematical and sequential reasoning benchmarks. Although the introduction and experimental setup mention commonsense reasoning, the main experiments do not include evaluations on such tasks. Despite using StrategyQA in Section 3.5 to demonstrate that RESPROMPT is not essential for simple questions, it would be valuable to showcase the effectiveness of RESPROMPT on more complex commonsense reasoning tasks such as CommonsenseQA or HotpotQA. Including experiments on these tasks would provide a comprehensive assessment of the efficacy of RESPROMPT across different domains of reasoning.

3.In section 3.3, it is mentioned that RESPROMPT demonstrates improved stability with variations in the number of exemplars in the prompt. However, it is difficult to observe from Figure 6 that RESPROMPT is more stable than other baselines and may even be less stable than the standard CoT. The results presented in the figure do not clearly indicate the claimed stability advantage of RESPROMPT.

**Questions:**

1.In section 3.3, it is mentioned that using the same tokens instead of symbolic variables yields better results. How would you explain this phenomenon? Intuitively, it seems challenging for the language model to reference the same tokens in the intermediate steps when they are not explicitly shown in the prompt.

2.This approach appears to somewhat decrease the readability of the reasoning process. What advantages does it offer in practical interactions with human?

3.How does RESPROMPT make the model aware of the graph structure during reasoning? Or does this linear modification in the prompt truly enable the model to learn the graph structure?

---

> ### Author Response · Authors · 2023-11-22
> **Response to Reviewer w7DW - Part 1**
>
> Dear Reviewer w7DW,
>
> We sincerely appreciate your positive comments and really thank you for the in-depth and insightful suggestions and questions! We provide a response one by one as follows. We hope these new results and explanation could address your concerns and we welcome further discussion.
>
>
> > **W1: Compare to more enhanced baselines for multi-step reasoning, and lack comparison with tree of thought and graph of thought.**
>
> **A1:** We greatly appreciate the reviewer for providing this feedback. We conduct new experiments on GSM8K to compare ResPrompt with advanced prompting methods for multi-step reasoning as described in [1], [2], and [3]. The results, presented in the following table, consistently demonstrate that ResPrompt outperforms these advanced baselines for multi-step reasoning. In the revised paper, we include a new section to incorporate these new comparison results and provide a detailed discussion.
>
>
> || DECOMP (1-shot) [1] | ResPrompt (1-shot) | Complexity (8-shot) [2] |ResPrompt (8-shot) | Least_to_most (4-shot) [3] | ResPrompt (4-shot) |
> |:-----------:|:----------:|:------------------------:|:---------:|:---------:|:---------:|:---------:|
> |LLaMA-65B | 40.4       | 46.6                   |48.3   |59.2 | 53.6 |58.4|
> |LLaMA2-70B| 50.3      | 57.2                   | 64.2  |65.3 | 60.1 |67.5 |
>
> Note that to ensure a fair comparison, we directly use the prompts as originally specified in the respective papers or open-sourced repositories [4] from the authors [2] of each baseline. Accordingly, for ResPrompt, we employ the same number of exemplars (x-shot) as used in each of these sourced prompts.
>
> Regarding the comparison with Tree of Thoughts (ToT) [5] and Graph of Thoughts (GoT) [6], we would like to clarify that ToT and GoT are more relevant for tasks that require strategic reasoning, such as backtracking, traversal, sorting, etc. The demo applications in [5, 6] include examples like sorting, document merging, game of 24, etc. On the other hand, ResPrompt aims to capture the complex underlying structure in standard multi-step problems. In summary, although both ResPrompt and ToT/GoT are related to the complex "structure”, ResPrompt targets different purposes compared to ToT and GoT. We include a more detailed discussion on this topic in the revised paper.
>
>
> > **W2: Run experiments on CommonsenseQA (CSQA) or HotpotQA. Including experiments on these tasks would provide a comprehensive assessment.**
>
> **A2:** We sincerely appreciate the reviewer for providing this valuable suggestion. We agree that including experimental results on these two benchmarks, which assess the knowledge ability of LLMs, would make the experiment more comprehensive and provide further insights into when ResPrompt may not be essential. We have included the results on CSQA and HotpotQA in the following table.
>
>
> | | CSQA (4-shot) | HotpotQA (4-shot) EM/F1 |
> |:-----------:|:----------:|:------------------------:|
> Standard| 78.6       |       33.3/43.2             |
> CoT | 79.0      |      37.3/50.9                   |
> ResPrompt| 78.0      | 36.3/49.9                  |
>
> These results demonstrate that ResPrompt can just achieve comparable performance to the baselines on both CSQA and HotpotQA. This outcome is not surprising since both benchmarks primarily require extensive knowledge to answer the questions, rather than complex multi-step reasoning. Therefore, it is natural that ResPrompt may not be essential in these knowledge assessment benchmarks. Due to time and resource limitations, we randomly selected 300 examples from each benchmark for these experiments.  We will include experiments on the full benchmarks in the revised paper.

---

> > ### Author Response · Authors · 2023-11-22
> > **Response to Reviewer w7DW - Part 2**
> >
> > > **W3: The stability of ResPrompt with the number of exemplars in Figure 6.**
> >
> > **A3:** We really appreciate the reviewer for these detailed comments. This is an excellent question, and we acknowledge that ResPrompt may not always exhibit stability with different numbers of exemplars, particularly in the AQUA-RAT dataset. We believe there are two possible reasons for this instability in AQUA-RAT:
> >
> > -  Unlike GSM8K, where most questions follow a similar math word format, AQUA-RAT has a more diverse type of question. It includes standard math word problems as well as more formal questions, such as "If m > 0, y > 0, and x is m percent of 4y, then, in terms of y, m is what percentage of x? A)y/400, B)4y, C)50y, D)2500/y, E)5000/y". Consequently, it becomes more challenging for the few-shot exemplars to cover all the diverse question types in AQUA-RAT. For example, if the 1-shot exemplar is a pure math word problem, LLMs may adhere to the math word problem format and are more likely to make mistakes when encountering the more formal question type mentioned above. This is particularly relevant to ResPrompt because its problem-solving process is more complex than that of CoT. LLMs may become stick to on the intricate reasoning process and, as a result, are less likely to solve other question types.
> >
> > - The test set of AQUA-RAT only has 254 examples. Compared to other benchmarks that mostly have 1000+ examples, the performance of AQUA-RAT may be more sensitive to different few-shot settings, especially for ResPrompt where the reasoning process in exemplars is more complex.
> >
> >
> >
> >
> > > **W4: Explanation of using same tokens for reference yields better results than symbolic variables.**
> >
> > **A4:** We really appreciate this in-depth question. We believe the reason is that using symbolic notation, such as "X," increases the reasoning difficulty for LLMs. Previous work [5] has shown that LLMs find it more challenging to understand symbolic notation compared to understanding semantics. By using the same tokens for reference, we essentially leverage the semantics within the problem-solving context, making it easier for LLMs to establish residual connections.
> >
> >
> > > **W5: Decreased readability and advantages in practical interactions with human?.**
> >
> > **A5:** We appreciate the reviewer a lot for raising this valuable practical question. It is indeed that ResPrompt may decrease the readability of the generated text due to its more detailed problem-solving process. However, we believe that the advantage of ResPrompt lies in its improved ability to solve complex questions that require multi-step reasoning for human users, such as complex mathematical problems. ResPrompt captures the interdependence between steps and derives a more accurate reasoning process and final result compared to existing approaches like CoT.
> >
> >
> > > **W6: How does ResPrompt make the model aware of the graph structure during reasoning?.**
> >
> > **A6:** We sincerely thank the reviewer for this insightful question. We would like to address this question using the prompt example in Figure 2. When comparing the reasoning flow of CoT (Figure 2 (c)) with the underlying reasoning graph of the question (Figure 2 (d)), it shows that the linear structure in Figure 2 (c) misses several links necessary to reconstruct the graph structure in Figure 2 (d). ResPrompt is designed to add these missing links. Although our modification builds upon the linear prompt of CoT, ResPrompt guides the model to realize the necessary intermediate results in earlier steps by repeating them. This repeating operation effectively adds the missing links and helps the model become aware of the graph structure.
> >
> > \
> > \
> > [1] Denny Zhou, Nathanael Schärli, Le Hou, Jason Wei, Nathan Scales, Xuezhi Wang, Dale Schuurmans, Claire Cui, Olivier Bousquet, Quoc V. Le, and Ed H. Chi. Least-to-most prompting enables complex reasoning in large language models. In  ICLR 2023,
> > [2] Yao Fu, Hao Peng, Ashish Sabharwal, Peter Clark, and Tushar Khot. Complexity-based prompting for multi-step reasoning. In ICLR 2023
> > [3] Tushar Khot, and Harsh Trivedi, Matthew Finlayson, Yao Fu, Kyle Richardson, Peter Clark, Ashish Sabharwal, Decomposed Prompting: A Modular Approach for Solving Complex Tasks, ICLR 2023
> > [4] https://github.com/FranxYao/chain-of-thought-hub
> > [5] Xiaojuan Tang, Zilong Zheng, Jiaqi Li, Fanxu Meng, Song-Chun Zhu, Yitao Liang, and Muhan Zhang. Large language models are in-context semantic reasoners rather than symbolic reasoner, 2023

---

### Official Review · Reviewer_n8nq · 2023-11-01

**Soundness:** 1 poor
**Presentation:** 2 fair
**Contribution:** 1 poor
**Rating:** 3
**Confidence:** 5

**Summary:**

This paper proposes a new prompting pipeline named ResPrompt to boost the performance of the LLMs, particularly focusing on complex reasoning tasks. The authors claim that the complex interdependence of prior reasoning steps during reasoning makes the question more challenging for LLMs. To alleviate this question, the authors propose to explicitly connect different reasoning steps and recap the LLM with prior reasoning results by repeating them in chain-of-thoughts.  Empirical evaluations demonstrate the potential of the proposed method.

**Strengths:**

The proposed method is neat and easy to follow. The authors provide relatively sufficient experiments to empirically demonstrate the performance improvement of the proposed method.

**Weaknesses:**

1. **Lacks sufficient verification of the necessity of the proposed method.** The authors claim that the complex interdependence between the current reasoning step and prior steps requires explicit connections among different reasoning steps. However, this is only an intuitive hypothesis and the authors do not verify whether it hold or not. Instead, they directly propose "repeating can help build the connections among reasoning steps and improve the performance". A better way to formulate the method could be designing simple yet straightforward experiments to verify whether the claim holds or not. For example, the authors can examine whether LLMs can make wrong references during a specific intermediate step. Consider the following CoT:

   ....So the money he earned from shoveling driveways is $110 - $15 - $60 = $35...

   We can change the above sentence into `So the money he earned is $110 - $15 - $60 = $35.` and then query the LLM again asking where the `$35` is earned from. If the model cannot correctly answer `from shoveling driveways`, then that means the authors' claim might hold as the model does make mistakes to recovering the references from prior reasoning steps.

2.  **Comparison with baselines might be unfair.** The authors claim that the performance can be achieved by more than 20% (relatively) on some datasets and models. However, if we look at the experiment setting and details of baselines, it seems the performance of baselines is suppressed, leading to a misunderstanding of the performance improvement. Specifically:

   a. There is no need to include the performance using the prompt used in [1], as the performance is too low.

   b. I do not understand why long prompts (`Long CoT`) lead to worse performance on the GSM8K dataset compared to short prompts (`Short CoT`), as the previous study [2] has already demonstrated that longer prompts provide better performance. The authors need clarifications and explanations for this phenomenon.

   c. It does not make sense to include questions with different numbers of reasoning steps in the `Long CoT`. Previous work [2] has already shown that using the most complex CoTs leads to the best performance. The author should include the original prompts used by [2] as another baseline. Note that the proposed method, due to repeating previous steps, uses significantly long prompts compared to baselines. Without including baselines that use prompts with similar lengths, it cannot be answered whether the performance improvement comes from longer prompts or the proposed method (repeating to connect intermediate steps)

   d. LLaMA performance on GSM8K could be much higher than the numbers in Table 1. See https://opencompass.org.cn/leaderboard-llm for a reference, where the LLaMA-2 achieves 63.5% accuracy and the LLaMA achieves 54.5% accuracy on GSM8K without any special techniques but CoTs. If we take this into consideration, the performance improvement is rather small.

3. **Cost-performance tradeoff is not discussed.** Again, the proposed method uses much longer CoTs compared to baselines. The author should discuss the inference cost given the long prompts.

4. **Additional ablation study.** I would like to emphasize this point again. The performance improvement could come from two aspects: longer prompts and proposed connections among different reasoning steps. The author should design an ablation study to demonstrate where the performance improvement comes from, rather than simply study which experiment designs can affect the final performance.

[1] Jason Wei, Xuezhi Wang, Dale Schuurmans, Maarten Bosma, Brian Ichter, Fei Xia, Ed H. Chi, Quoc V. Le, and Denny Zhou. Chain-of-thought prompting elicits reasoning in large language models. In NeurIPS, 2022b.

[2] Yao Fu, Hao Peng, Ashish Sabharwal, Peter Clark, and Tushar Khot. Complexity-based prompting for multi-step reasoning. In ICLR 2023

**Questions:**

Please refer to the weakness above. Although I give a low rating to this paper, I would be delighted to increase my rating given my questions addressed.

---

> ### Author Response · Authors · 2023-11-22
> **Response to Reviewer n8nq -Part 1**
>
> Dear Reviewer n8nq,
>
> We sincerely appreciate your detailed and insightful comments. We provide explanations to your concerns point-by-point in the following. We hope that these new results and explanations adequately address your concerns, and we welcome further discussion.
>
>
> > **W1: Lacks sufficient verification of the necessity of the proposed method.**
>
> **A1:** We greatly appreciate the reviewer for the valuable feedback and the insightful experiment suggestion. We conduct a quick verification experiment using LLaMA2-70B as per the suggestion. However, upon further consideration, we realize that the reviewer's suggested query, "query the LLM again asking where the ``$35`` is earned from," may not align exactly with our original motivation. This is because the amount of ``$35`` is derived from the immediately previous step, whereas our main observation is that the model struggles to connect with "results of several steps earlier". Therefore, instead of querying the LLM about the source of ``$35``, we modify the query to ask where ``$110`` is earned from. Specifically, we used the following prompt to query LLaMA2-70B:
>
> “Question: Tobias is buying a new pair of shoes that costs ``$95``. He has been saving up his money each month for the past three months. He gets an ``$5`` allowance a month. He also mows lawns and shovels driveways. He charges ``$15`` to mow a lawn and ``$7`` to shovel. After buying the shoes, he has ``$15`` in change. If he mows 4 lawns, how many driveways did he shovel? Before the purchase, he had ``$95 + $15 = $110`` in total. From his allowance, he earned ``3 * $5  = $15``. From mowing lawns, he earned ``4 * $15 = $60``. Where the ``$110`` is earned from?”
>
> The greedy decoding output of LLaMA2-70B base model is
> “``$15 + $60 = $75``. From shoveling driveways, he earned ``$110 - $75 = $35``. How many driveways did he shovel? ``$35 / $7 = 5``. He shoveled 5 driveways.”, which continues to solve the problem but does not answer the question. So we use a softer temperature and query LLaMA2-70B chat model for 5 times. The outputs are shown in following
>
> 1. Tobias earned ``$110`` from mowing lawns and shoveling driveways combined. He mowed 4 lawns, so he shoveled ``($110 - $60) / $7 = 12`` driveways.
> 2. The ``$110`` is earned from the sum of the money earned from his allowance and the money earned from mowing lawns.
> 3. Tobias earned ``$110`` from his allowance and mowing lawns combined. He earned ``$15`` from his allowance, and ``$60`` from mowing lawns.
> 4. The ``$110`` Tobias earned came from his allowance and mowing lawns. He earned ``$15`` from his allowance and ``$60`` from mowing lawns, totaling ``$110``.
> 5. Tobias earned the ``$110`` from his allowance and mowing lawns. He earned ``$15`` from his allowance and ``$60`` from mowing lawns, which is a total of ``$75``. Where the remaining ``$35`` came from is not specified in the information provided.
>
> From these results, we notice that LLaMA2-70B can not correctly answer the source of ``$110``. We believe this could be an empirical verification that supports our claim and motivation.
>
>
>
> > **W2:Comparison with baselines might be unfair**
>
> > **W2 (a) : There is no need to include the performance using the prompt used in [1], as the performance is too low.**
>
> **A2 (a):** We thank the reviewer for this detailed comment. We agree that the prompts used in [1] are relatively simple. However, considering that the LLaMA family of models, particularly LLaMA2, is relatively new, we believe that including the performance of LLaMA2 with these prompts could provide a reference for the community. Additionally, in certain experiments, such as AQUA-RAT, the original prompts can still serve as strong baselines. For these reasons, we have decided to include these prompts in our evaluation.

---

> > ### Author Response · Authors · 2023-11-22
> > **Response to Reviewer n8nq -Part 2**
> >
> > > **W2 (b) : Why long prompts (Long CoT) lead to worse performance on the GSM8K dataset compared to short prompts (Short CoT), which is different from  [2].**
> >
> > **A2 (b):** We sincerely appreciate the reviewer for raising this question, and we would like to clarify that the "Long CoT" in our paper does not refer to the "complexity-based prompts" mentioned in [2]. In [2], "complexity-based prompts" involve selecting examples with higher complexity, such as a higher number of reasoning steps per example. This approach falls under the category of **better example selection**, as stated in [2]. Specifically, according to the open-source repository [4] provided by the authors of [2], they select questions with long steps (mostly 8 or 9 steps) for prompting. We would like to highlight that our work does not focus on example selection but rather on **improving the problem-solving process in prompts** through the use of residual connections. Therefore, the examples we use to design prompts are different from those used in either [1] or [2]. To ensure a fair comparison and to ensure that our improvements stem from the residual connections rather than different example selection, we use the same examples to derive CoT prompts, following the same operations for fair comparison as described in [3]. As our examples generally have a greater number of steps compared to [1], we refer to this baseline as "Long CoT".
> >
> > In summary, the term "Long CoT" does not refer to the complexity-based prompts in [2]. We also add the experiments of comparing with [2] in the response of the next question in the following.  However, we acknowledge that using the terms "Long CoT" and "Short CoT" may cause confusion, and we revise them to "Derived CoT" and "Original CoT" in our revised version.
> >
> >
> > > **W2 (c) : The author should include the original prompts used by [2] as another baseline.**
> >
> > **A2 (c):** We sincerely appreciate the reviewer for this experiment suggestion. We run new experiments using two versions of the original prompts in [2] with the highest complexity on GSM8K. These versions include prompts with 1) all 8 steps or 2) half 9 steps and half 8 steps, both are in the 8-shot setting. The results are presented in the following table. We observe that ResPrompt consistently outperforms these baselines. We include these results in our revised version.
> >
> >
> > || All 8-Step | Half 9-Step & Half 8-Step | ResPrompt |
> > |:-----------:|:----------:|:------------------------:|:---------:|
> > |LLaMA1-65B| 48.3      | 49.6                   | 58.4  |
> > |LLaMA2-70B| 64.2      | 63.8                   | 65.3  |
> >
> > We would like to highlight, as mentioned in b), that the complexity based prompts in [2] aim to select better examples, while our ResPrompt aims to improve the problem-solving process in prompts. These are **two different perspectives** to enhance multi-step reasoning prompts. Therefore, our "Long CoT" baseline (again, we will change it to “Derived CoT”) is not another implementation of the complexity-based prompts in [2]. Instead, it is derived from ResPrompt using the same examples to demonstrate that the improvement stems from **building residual connections** rather than solely using different examples compared to CoT in [1].
> >
> >
> > > **W2 (d) : Higher LLaMA CoT performance on GSM8K, see https://opencompass.org.cn/leaderboard-llm for a reference.**
> >
> > **A2 (d):** We sincerely appreciate the reviewer for providing a valuable reference to this wonderful leaderboard. Upon checking the GSM8K prompts used in the leaderboard, we notice that they are 4-shot complexity-based prompts. Specifically, the leaderboard utilizes 4 GSM8K exemplars selected by [2], with each exemplar consisting of 9 reasoning steps of problem-solving. Therefore, we would like to highlight that these results are obtained through principled complexity-based example selection.
> >
> > To compare with these results, we conduct 4-shot experiments using our ResPrompt. The results, shown in the following table, indicate that ResPrompt still achieves better performance than the leaderboard in 4-shot setting. We include these results in our revised version.
> >
> > || Leaderboard (9-Step) | ResPrompt |
> > |:-----------:|:----------:|:---------:|
> > |LLaMA1-65B| 54.5      |  59.2  |
> > |LLaMA2-70B| 63.5       |  67.5  |

---

> > > ### Author Response · Authors · 2023-11-22
> > > **Response to Reviewer n8nq -Part 3**
> > >
> > > > **W3 : Cost-performance tradeoff is not discussed.**
> > >
> > > **A3:** We really thank the reviewer for this feedback. We have done a new analysis on GSM8K for the cost-performance study. We run these experiments using LLaMA2-70B in the 8-shot setting. For the complexity based prompts, we follow the repo in [4]  The results are shown in the following table.
> > >
> > > || Number of Tokens | Inference Speed | Performance|
> > > |:-----------:|:----------:|:---------:|:---------:|
> > > |Original CoT| 1      |  1  | 57.3 |
> > > |Complexity| 3.76X       |  0.56X  | 64.2 |
> > > |ResPrompt| 3.06X       |  0.65X  | 65.3 |
> > >
> > > On average, the combined tokens of prompts and outputs of ResPrompt is ~3.06X more than the tokens in the original CoT on the entire GSM8K test set, while the inference speed of ResPrompt is ~0.65X of the original CoT. We acknowledge that our prompt is longer than the original CoT and thus has higher inference cost. However, ResPrompt only has 3.06X/3.76X = 0.81X tokens and is 0.65X/0.56X = 1.16X faster than the complexity based in terms of inference speed, while achieving a better performance. We include this discussion in our revised paper to help the reader better understand our approach.
> > >
> > >
> > >
> > > > **W4 : Additional ablation study.**
> > >
> > > **A4:** We sincerely appreciate this suggestion, and we believe that including these results will greatly enhance the comprehensiveness of our paper. Here we would like to discuss two ablation studies.
> > >
> > >
> > > 1.  To ensure that the observed improvement stems from building residual connections rather than longer prompts, as mentioned in W2 (c) and W2 (d), we compare our ResPrompt with the complexity-based prompting approach in [2], which utilizes very long prompts. Specifically, the length of the complexity-based prompts is approximately 1.16 times longer than ResPrompts in terms of token count. We present the results in the following again for easy reading:
> > >
> > > - 8-shot:
> > > || All 8-Step | Half 9-Step & Half 8-Step | ResPrompt |
> > > |:-----------:|:----------:|:------------------------:|:---------:|
> > > |LLaMA1-65B| 48.3      | 49.6                   | 58.4  |
> > > |LLaMA2-70B| 64.2      | 63.8                   | 65.3  |
> > >
> > >
> > > - 4-shot:
> > > || Leaderboard (9-Step) | ResPrompt |
> > > |:-----------:|:----------:|:---------:|
> > > |LLaMA1-65B| 54.5      |  59.2  |
> > > |LLaMA2-70B| 63.5       |  67.5  |
> > >
> > > Compared to different versions of long prompting in [2], ResPrompt consistently achieves better reasoning performance. So we can conclude that the improvement comes from **building residual connections instead of longer prompts**.
> > >
> > > \
> > > 2. We would also like to highlight another ablation study already in our paper, which compares ResPrompt with Long-CoT (again, will change to Derived-CoT) in Table 1. Long CoT (Derived-CoT) refers to the CoT prompts derived from the same examples used in ResPrompt. This ablation study ensures a fair comparison and validates that our improvements stem from **residual connections rather than solely from using different data examples to design prompts**.
> > >
> > > We include and highlight these results in the revised paper.
> > >
> > >
> > >
> > > \
> > > \
> > > [1] Jason Wei, Xuezhi Wang, Dale Schuurmans, Maarten Bosma, Brian Ichter, Fei Xia, Ed H. Chi, Quoc V. Le, and Denny Zhou. Chain-of-thought prompting elicits reasoning in large language models. In NeurIPS, 2022b.
> > > [2] Yao Fu, Hao Peng, Ashish Sabharwal, Peter Clark, and Tushar Khot. Complexity-based prompting for multi-step reasoning. In ICLR 2023.
> > > [3] Denny Zhou, Nathanael Schärli, Le Hou, Jason Wei, Nathan Scales, Xuezhi Wang, Dale Schuurmans, Claire Cui, Olivier Bousquet, Quoc V. Le, and Ed H. Chi. Least-to-most prompting enables complex reasoning in large language models. In  ICLR 2023,
> > > [4] https://github.com/FranxYao/chain-of-thought-hub.

---

> > > > ### Comment · Reviewer_n8nq · 2023-12-02
> > > >
> > > > I really appreciate the authors for the detailed response. Here are my comments about the response:
> > > >
> > > > 1. A2(a): However, considering that the LLaMA family of models, particularly LLaMA2, is relatively new, we believe that including the performance of LLaMA2 with these prompts could provide a reference for the community.
> > > >
> > > > It would be great to include the weaker method for comparison. But it is not acceptable to use those weaker method to demonsrate the superiority of your method.
> > > >
> > > > 2. A2(c): comparision with Complex-CoT
> > > >
> > > > The experiment results further verify my hypothesis, where given longer and more complex annotations, there is no need to adopt the proposed method.
> > > >
> > > > Also, as pointed out by other reviewers (and verified by the authors), the performance gain of the proposed method is more subtle when equipped with more powerful models such as GPT-4. In summary, I appreciate the authors' effort but still cannot support the acceptance of this paper.

---

### Author Response · Authors · 2023-11-22
**To All Reviewers**

Dear Reviewers:

First of all, sorry for the delayed response, as we are busy preparing the additional experiments. We would like to express our sincere appreciation to all reviewers for their valuable comments and insightful suggestions. We have revised our paper as suggested by reviewers. We highlight our major changes using **red** color text in the revised PDF. We also summarize the major modifications as follows:

1. We change the terminology of “Short CoT” to “Original CoT”, and “Long CoT” to “Derived CoT” throughout section 3 (the experiment section), to avoid the confusion between “Long CoT” and complexity based prompting [1] (per Reviewer n8nq, Reviewer z7DW and Reviewer z49G)

2. We add experiment comparison of ResPrompt with the original prompts of complexity based prompting in [1], as suggested by Reviewer n8nq. Due to page limit, this can be found in Appendix section D.8

3. We include experimental discussion about the cost-performance analysis of ResPromt and baselines, as suggested by Reviewer n8nq. Due to page limit, this can also be found in Appendix section D.8

4. We include the new ablation about long prompts  (the same experiment in 2) in Appendix section D.8 and further discuss  the ablation to different examples in prompts in section 3. This is also suggested by Reviewer n8nq.

5. We add the new experiments of ResPrompt compared with more advanced multi-step prompting approaches [2,3] as suggested by Reviewer w7DW, This can be found  in Appendix section D.8,

6. We add new experiments on CSQA and HotpotQA datasets to understand whether it is essential to build residual connections for reasoning tasks requiring extensive knowledge, as suggested by Reviewer w7DW. This result can be found in Appendix section D.9.

7. We add new experiments using self-consistency [4] in Appendix section D.8. This is suggested by Reviewer z49G.

8. We add new experiments using more powerful LLMs, including  GPT-3.5 and GPT-4,   in Appendix section D.8. This is also suggested by Reviewer z49G.


\
\
[1] Yao Fu, Hao Peng, Ashish Sabharwal, Peter Clark, and Tushar Khot. Complexity-based prompting for multi-step reasoning. In ICLR 2023
[2] Tushar Khot, and Harsh Trivedi, Matthew Finlayson, Yao Fu, Kyle Richardson, Peter Clark, Ashish Sabharwal, Decomposed Prompting: A Modular Approach for Solving Complex Tasks, ICLR 2023
[3] Denny Zhou, Nathanael Schärli, Le Hou, Jason Wei, Nathan Scales, Xuezhi Wang, Dale Schuurmans, Claire Cui, Olivier Bousquet, Quoc V. Le, and Ed H. Chi. Least-to-most prompting enables complex reasoning in large language models. In  ICLR 2023,
[4] Xuezhi Wang, Jason Wei, Dale Schuurmans, Quoc Le, Ed Chi, Sharan Narang, Aakanksha Chowdhery, Denny Zhou. Self-consistency improves chain of thought reasoning in language models ICLR 2023.

---

> ### Author Response · Authors · 2023-11-22
> **Look forward to the discussion**
>
> Dear Reviewers,
>
> We sincerely appreciate the time and effort you've devoted to reviewing our work. We understand that your schedule may be quite busy. As the authors-reviewer discussion phase draws to a close, we kindly request your attention to our response. Our aim is to gain insights into whether our response effectively addresses your concerns and to ascertain if there are any additional questions or points you would like to discuss.
>
> We look forward to the opportunity for further discussion with you. Thank you for your thoughtful consideration.
>
> Best regards, \
> Authors